# Automated Label Unification for Multi-Dataset Semantic Segmentation with GNNs

**Rong Ma**,* **Jie Chen**,* **Xiangyang Xue, and Jian Pu**†
Fudan University
rma22@m.fudan.edu.cn, {chenj19,xyxue,jianpu}@fudan.edu.cn

## Abstract

Deep supervised models possess significant capability to assimilate extensive training data, thereby presenting an opportunity to enhance model performance through training on multiple datasets. However, conflicts arising from different label spaces among datasets may adversely affect model performance. In this paper, we propose a novel approach to automatically construct a unified label space across multiple datasets using graph neural networks. This enables semantic segmentation models to be trained simultaneously on multiple datasets, resulting in performance improvements. Unlike existing methods, our approach facilitates seamless training without the need for additional manual reannotation or taxonomy reconciliation. This significantly enhances the efficiency and effectiveness of multi-dataset segmentation model training. The results demonstrate that our method significantly outperforms other multi-dataset training methods when trained on seven datasets simultaneously, and achieves state-of-the-art performance on the WildDash 2 benchmark. Our code can be found in https://github.com/Mrhonor/AutoUniSeg.

## 1   Introduction

Recent advances in computer vision [35, 21] have shown the advantages of large datasets in training robust visual models [2]. However, for deep supervised visual models that rely on annotated data, the collection of such extensive annotated datasets can be prohibitively costly [11]. To address this expense and expand the data available for training, several efforts [4, 23, 30] focus on the challenges of multi-dataset training, enabling the use of diverse datasets to train more robust and generalizable models.

Models trained on multiple datasets must confront the challenge of reconciling conflicting annotation standards and label spaces. For example, the class *road* in the BDD dataset [46] can be further divided into several classes in the Mapillary dataset [33]: *road*, *lane marking* and *crosswalk*. Similarly, the Mapillary dataset labels both *barrier* and *curb* as distinct classes, while in the IDD dataset [42], they are combined under the single label *curb*. These conflicts impact the supervised learning of models, as they may be incorrectly penalized for predicting finer-grained classes from other datasets.

Another challenge is the task of unifying diverse dataset labels to produce outputs in a standardized format [29, 24]. Several methods [11, 30] attempt to address this by concatenating the label spaces of all datasets and using language models to encode label names into a text embedding space. However, there approaches introduce redundancy and fail to handle issues where labels share names but differ in annotation granularity. Other methods involve manually constructing universal taxonomies [4, 24] or relabeling [23], both of which are time-consuming and labor-intensive. Recent approaches aim to

---

*These authors contributed to the work equally and should be regarded as co-first authors.
†Corresponding author.

38th Conference on Neural Information Processing Systems (NeurIPS 2024).

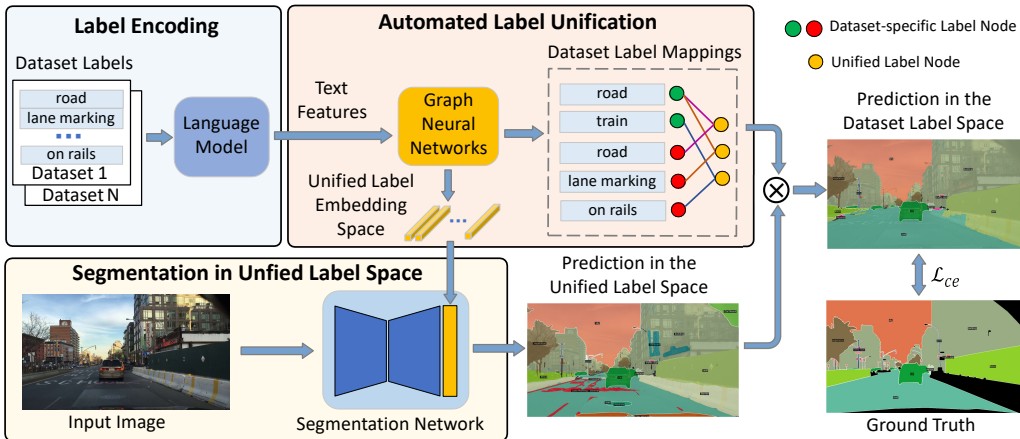

Figure 1: Our method consists of three modules. The label encoding provides the semantic text features of the dataset labels. The GNNs learn the unified label embedding space and dataset label mappings based on the textual features and input images. The segmentation network leverages the unified label embedding space to produce segmentation results in the unified label space.

automatically construct universal taxonomies [6, 41], but these methods typically identify inter-label relations between only two datasets, involving a time-consuming iterative training process.

In this paper, we propose a novel approach leveraging Graph Neural Networks (GNNs) [20] to automatically construct unified label space, enabling segmentation model to be trained simultaneously on multiple datasets. In contrast to previous approaches [11, 4, 6], our approach eliminates the need for manual re-annotation or iterative training procedures to construct universal taxonomies, while also addressing the limitation of language-based methods in distinguishing categories with identical semantics. As depicted in Figure 1, we utilize a language model to convert the dataset labels into text features. Then, we apply GNNs to learn the relationships and associations among these labels. The process creates a unified label embedding space and dataset label mappings. The output head of the segmentation network incorporates the unified label embedding space to generate unified segmentation results within this unified space. The dataset label mappings are subsequently utilized to align the unified segmentation results with the label spaces relevant to each dataset. This enables the training of segmentation models and graph neural networks with dataset-specific annotations.

## 2   Related work

**Multi-datasets Semantic Segmentation.** In recent years, numerous studies [52, 29] focused on training semantic segmentation models on multiple datasets. A simple approach involved incorporating dataset-specific modules in the model, such as dataset-specific output heads [28] or dataset-specific batch normalization layers [44], to produce predictions tailored to each dataset domain. While these methods effectively avoided issues of dataset label space conflicts, they offered limited applicability in real-world scenarios as they could not deliver unified predictions. MSeg [23] addressed the problem at the data level via manual re-annotation processes to resolve label conflicts. However, this approach was time-consuming, error-prone and not easily scalable. Recent methods employed manual [5] or automatic techniques [6] to construct universal taxonomies and establish label relationships between datasets label space and the unified label space. These methods, leveraging partial label learning approaches [48, 12], enabled training with dataset-specific annotations and producing unified predictions.

**Construct Universal Taxonomies.** Each dataset has its unique domain, necessitating the construction of universal taxonomies to enable the model to cover all domains. Several approaches [11, 30] attempted to concatenate dataset label spaces and differentiated similar classes by aligning text embedding encoded by language model. However, this approach struggled with classes that had the same name but different levels of annotation granularity, and direct concatenation of label spaces could lead to semantic conflicts. Other research [24, 29] attempted to establish a unified label space

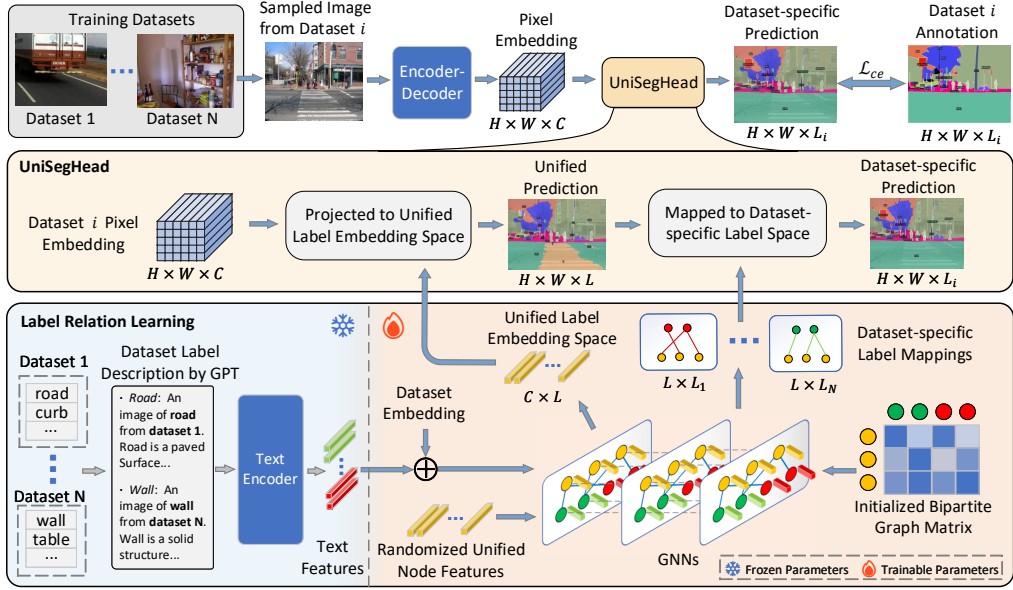

Figure 2: Illustration of our method that training with dataset-specific annotations through label mappings constructed by GNNs. We leverage a unified segmentation head (UniSegHead) to enable simultaneous training on multiple datasets. In the UniSegHead, we compute the matrix product between pixel embedding and augmented unified node features output by the GNNs, resulting in predictions for the unified label space. We finally utilize the label mappings constructed by GNNs to map the unified predictions to dataset-specific prediction for training.

through the expertise of human annotators. Additionally, other studies [41, 53] aimed to address this issue by developing automated mechanisms to create universal taxonomies. However, these methods either could only construct a unified label space between two datasets at a time [6] or could not handle complex class relationships [53]. Our approach stands out by automatically constructing universal taxonomies in a single training session with multiple datasets, resulting in significant time savings compared to methods that require multiple training iterations.

**Graph Neural Networks** demonstrated exceptional effectiveness in dealing with complex topological data structures, as highlighted in recent research [43]. The applicability extended across various domains, including recommendation systems [18], knowledge graph construction [50], and skeletal action recognition [54]. In the context of our problem, discovering label relations can be conceptualized as a link prediction task [49, 22]. However, conventional approaches for graph link prediction [9, 10] are not suitable for our model since we do not possess ground truth links. We use the values of the learnable adjacency matrix as predictions of whether nodes are linked. Supervision of the linked predictions relys on the segmentation results of the segmentation network and the corresponding image annotations.

## 3 Proposed Method

The comprehensive framework is depicted in Figure 2. We first define the unified representation of the multi-dataset label space. Utilizing this representation, we build a graph neural network to learn the unified label space. Finally, leveraging the unified label space, we train our segmentation network and the graph neural network using the dataset annotations.

### 3.1 Unification of Multi-dataset Label Space

**Unified Label Space.** Given $K$ datasets with their respective label space $\{L_1, L_2, \ldots, L_K\}$, multi-dataset semantic segmentation requires a model to predict within a consistent label space that

encompasses all dataset label spaces $L_{pred} = \bigcup_{i=1}^{K} L_i$. Each pixel must be assigned to a label in this unified label space. We define $N$ unified label nodes $\mathcal{A} = \{\alpha_1, \alpha_2, \ldots, \alpha_N\}$, serving as the nodes in the graph neural networks. Their corresponding $D$-dimensional learnable embedding $\{\mathbf{x}_1, \mathbf{x}_2, \ldots, \mathbf{x}_N\}$ represent the unified label embedding space. The number of unified label nodes $N$ is often smaller than the total number of dataset classes $|L|$, because we aim to merge multiple identical classes into a single unified label. The image is first encoded into pixel embedding $\mathcal{P}$ by the segmentation network, which is then projected into the unified label embedding space to assign a unified label to each pixel.

**Label Mappings.** We define a mapping from the unified label space to the dataset-specific label space $\mathbf{M}_i : \mathcal{A} \rightarrow L_i$, which is used to train the model with dataset-specific annotations. Mathematically, $\mathbf{M}_i \in \{0,1\}^{N \times |L_i|}$ is a boolean linear transformation. Each unified class $\alpha$ is at most linked to a class $c$ within a specific dataset $i$ to prevent label conflicts: $\mathbf{M}_\alpha \mathbf{1} \leq \mathbf{1}$. To handle different annotation granularities, we use label mappings to merge multiple unified label nodes representing fine-grained classes into one super-class $c$: $\mathbf{M}_c^\top \mathbf{1} \geq \mathbf{1}$. For example, the *curb* from IDD can be simultaneously mapped by unified label nodes represented *curb* and *barrier*. We use unified label nodes as input nodes for the GNNs, which learn the label mappings and unified label embedding space, thus enabling the automated unification of multi-dataset label spaces.

## 3.2 Learning Unified Label Space with Graph Neural Networks

Learning the label mappings between dataset label spaces and the unified label space can be viewed as a bipartite graph matching problem. This makes graph neural networks well-suited to address this issue. Below, we detail our approach of constructing GNNs for learning a unified label space.

**Input Nodes.** To construct the input feature of dataset-specific label nodes, as illustrated in Figure 2, we used the dataset labels in the template "An image of <label> from the dataset <dataset>" as plain text input and employed ChatGPT to complete the detailed description of each label. Then, we employ llama-2 [40] to encode these label descriptions and generate text features. To further distinguish nodes from different datasets, we introduce learnable dataset embedding for each dataset $\{\mathbf{d}_1, \mathbf{d}_2, \ldots, \mathbf{d}_K\}$. The dataset embedding is combined with the text features to form the input features for dataset-specific label nodes:

$$\mathbf{x}_{i,m} = f_t(l_{i,m}) + \mathbf{d}_i, \tag{1}$$

where $\mathbf{x}_{i,m}$ is the input feature of the $m$-th label from dataset $i$, and $f_t(l_{i,m})$ is the text feature of the label description $l_{i,m}$ encoded by language model. The input features of unified label nodes are randomly initialized to the same dimension as the dataset-specific label node. To determine the appropriate number of unified label nodes, inspired by the approach in [53], we used cross-validation results across different datasets to identify the number of mergeable categories, which served as the initial selection for the unified label nodes. The specific algorithm can be found in Appendix B. Together, dataset-specific label nodes and unified label nodes constitute the input nodes of GNNs. During the training process, we maintain a constant number of nodes. After the training is completed, we will remove inactive unified label nodes, meaning those that were not assigned to any dataset-specific label.

**Learnable Adjacency Matrix.** To enable label mappings to be updated via gradient descent, we embed the label mappings as a continuous, learnable graph adjacency matrix $\mathbf{M}_a$. Values in the adjacency matrix represent the weights of corresponding edges. Only the edges between unified label nodes and dataset-specific nodes are learnable, while others are fixed to zero. We apply a softmax operation to the edges connecting each unified label node with nodes of a particular dataset, ensuring that the sum of the edge weights $w$ between each unified label node and nodes of this dataset equals one. The element at the intersection of the $r$-th row and the $c$-th column in the lower triangular portion of $\mathbf{M}_a$ is formulated as:

$$\mathbf{M}_{r,c} = \begin{cases} \dfrac{e^{w_{r,c}}}{\sum_{c' \in L_i, r' \in \mathcal{A}} e^{w_{r',c'}}} & \text{if } r > |L| \text{ and } c < |L| \\ 0 & \text{otherwise.} \end{cases} \tag{2}$$

**Unified Label Embedding.** The forward propagation of our graph model follows the GraphSAGE framwork [17] as formulated in Equation 3. $\mathbf{W}^k$ and $\mathbf{X}^k$ are the weight and feature of the $k$-th GNNs layer. The $\sigma$ indicates nonlinear activation function, implemented as the tanh in this work. The output

Table 1: Training and test datasets in our experiments.

| Dataset Domain | Training and Validation datasets | Unseen test datasets |
|---|---|---|
| Driving scene | CityScapes [13], Mapillary [33], BDD [46], IDD [42] | WildDash 2 [47], KITTI [16], CamVid [8], |
| Indoor scene | SUN RGBD [37] | ScanNet-20 [14] |
| Everyday objects | ADE20K [51], COCO [25] | PASCAL VOC [15], PASCAL Context [32] |

features of the unified label nodes from the final layer serve as the unified label embedding space, $\mathbf{X}_u = [\mathbf{x}_1, \mathbf{x}_2, \ldots, \mathbf{x}_N]^\top$.

$$\mathbf{X}^{k+1} = \sigma(\mathbf{W}^k[\mathbf{X}^k \| \mathbf{M}_a \mathbf{X}^k]). \tag{3}$$

To obtain the dataset label mappings, we partition the adjacency matrix into submatrices, each corresponding to a specific dataset. Each submatrix contains only the unified label nodes and the label nodes specific to that dataset. We compute the label mappings for each dataset based on the values in its corresponding submatrix, divided into the following two cases. During training, we will alternately train the segmentation network and the GNNs. When training the GNNs, we directly use the value of the learnable adjacency matrix to establish label mappings, thereby facilitating weight updates through gradient descent. When training the segmentation network, we utilize the unbalanced optimal transport algorithm to solve for the boolean label mappings that satisfies the many-to-one mapping constraints. This algorithm detailed in Appendix C effectively handles the conversion of the continuous adjacency matrix into a discrete dataset label mappings required for training segmentation network.

## 3.3 Training a Universal Model with Dataset-specific Annotations

Training a universal model is divided into two steps. The first step involves training a robust encoder-decoder to provide the pixel embedding for each pixel position in the image, where similar objects should have similar features. The second step focuses on learning label mappings and unified label embedding space by GNNs. During the training phase, we alternate between these two steps, freezing one network while training another network. Both of these steps require supervised training using dataset-specific annotations. Here, we primarily focus on the training of GNNs, while further details of the training strategies are provided in the Appendix A.

**Training with Dataset-specific Annotations.** During training, an image is randomly sampled from dataset $i$ and fed into an segmentation network, which provides embedding for each pixel position $\mathcal{P} = \{\mathbf{p}_1, \mathbf{p}_2, \ldots, \mathbf{p}_j\}$, where $\mathbf{p}_k$ is $D$-dimensional vectors. We utilize a unified segmentation head (UniSegHead) to assign a dataset label to each pixel. In the UniSegHead, We first project the pixel embedding into the unified label embedding space by multiplying the pixel embedding by the output feature of unified label node $\mathbf{X}_u$, as shown in Equation 4. Then, to train the universal model with dataset-specific annotations, we need to map the predictions in unified label space to dataset-specific label space to obtain the probabilities of dataset-specific classes. This is achieved by computing per-pixel dataset-specific logits $\mathbf{s}$ by multiplying the dataset-specific label mappings $\mathbf{M}_i$ at each pixel:

$$\mathbf{u}_k = \mathbf{X}_u \mathbf{p}_k, \tag{4}$$

$$\mathbf{s}_k = \mathbf{M}_i \mathbf{u}_k. \tag{5}$$

Finally, probabilities of dataset-specific classes are computed by per-pixel softmax operation over the logits $s$. This allows us to use dataset-specific annotations to compute the pixel-wise loss function, train the network, and update the label mappings. We formulate the cross-entropy loss for a specific pixel to train the segmentation network:

$$\mathcal{L}_{ce}(\mathbf{y}, \mathbf{s}) = -\sum_{c=1}^{|L_i|} y_c \log(\mathrm{softmax}(\mathbf{s})_c), \tag{6}$$

where $\mathbf{y}$ is the pre-pixel annotation, $|L_i|$ represents the total number of classes in the dataset $i$ from which the image originates.

**Orthogonality Loss.** To achieve a conflict-free unified label space and avoid redundant unified label nodes that represent the same class or have overly similar features, we introduce soft constraints to promote orthogonality among the unified label node features, inspired by [39]. This orthogonality loss encourages unified label node embedding to be mutually orthogonal. It not only aligns the unified

Table 2: Multi-dataset performance compared with other methods.

| Methods | Backbone | Venue | Label space[1] | CS | MPL | SUN | BDD | IDD | ADE | COCO | Mean |
|---|---|---|---|---|---|---|---|---|---|---|---|
| MSeg [23] | HRNet-W48 | CVPR 20 | MR | 76.3 | 51.9[2] | 46.1 | 63.5 | 61.8 | 42.8[2] | 48.6[2] | 55.9 |
| NLL+ [4] | SNp-RN18 | WACV 22 | MC | 72.6 | 39.1 | 41.7 | 58.5 | 54.4[3] | 31.0 | 35.4 | 47.5 |
| Uni NLL+ [5] | SNp-DN161 | IJCV 24 | MC | 76.1 | **44.2** | 46.9 | 60.4 | 56.7[3] | 35.6 | 39.3 | 51.3 |
| Single dataset | HRNet-W48 | - | DS | 77.0 | 30.2 | 43.9 | 62.4 | 66.8 | 34.5 | 38.0 | 50.4 |
| Multi-SegHead | HRNet-W48 | - | DS | 79.5 | 36.1 | 47.3 | **65.6** | 67.0 | 30.5 | 36.7 | 51.8 |
| Auto univ. [6] | SNp-RN18 | BMVC 22 | Auto | 72.7 | 35.8 | 42.3 | 59.6 | 55.2[3] | 30.7 | 35.6 | 47.4 |
| Ours | HRNet-W48 | - | Auto | **80.7** | 43.7 | **47.5** | 65.5 | **68.6** | **42.0** | **46.7** | **56.4** |

[1] Approach to construct label space. MC:Manually Construct, MR:Manually Relabel, DS:Dataset-specific, Auto:Automatically Construct.

[2] MSeg train and evaluate on 43 of 65 class in Mapillary dataset, 117 of 150 class in ADE dataset, 122 of 133 class in COCO dataset.

[3] These methods were trained and evaluated using 30 classes from the IDD dataset, while we trained and evaluated using the officially recommended 26 classes.

Table 3: Performance comparison with two baselines on training and unseen datasets.

| Trained dataset or label space | Mean results across training datasets | | Mean results across unseen datasets | |
|---|---|---|---|---|
| | Single dataset | Multi-SegHead | Single dataset | Multi-SegHead |
| CS | 23.9 | 30.9 | 31.4 | 37.9 |
| MPL | 30.6 | 36.1 | 36.9 | 41.0 |
| SUN | 12.9 | 22.0 | 16.7 | 28.9 |
| BDD | 24.8 | 29.1 | 30.4 | 35.7 |
| IDD | 26.0 | 34.0 | 28.9 | 36.4 |
| ADE | 28.5 | 40.3 | 37.2 | 49.8 |
| COCO | 30.5 | 38.8 | 45.1 | 53.2 |
| Ours | **56.4** | | **56.9** | |

label nodes with annotation standards for practical use but also enhances the diversity of the model and helps in finding a better label mappings:

$$\mathcal{L}_{orth} = -\sum_{i=1}^{N} \text{softmax}(\mathbf{X}_u \mathbf{x}_i)_i \log(\text{softmax}(\mathbf{X}_u \mathbf{x}_i)_i). \tag{7}$$

The final loss function used to train the GNNs is represented as follows, with $\lambda$s as hyperparameters to adjust the weights of different loss functions:

$$\mathcal{L} = \lambda_1 \mathcal{L}_{ce} + \lambda_2 \mathcal{L}_{orth}. \tag{8}$$

## 4 Experiments

**Datasets.** We list the semantic segmentation datasets used for training and testing in Table 1. Our training datasets cover a wide range of scenarios, from indoor scenes to driving scenes. We also introduce corresponding test datasets, which are not used in the training process, for the respective scenes to evaluate our generalization capability.

**Implementaion Details.** Our segmentation model is based on the HRNet-W48 architecture [38], while the GNN model is a three-layer GraphSAGE [17]. We utilize the llama-2-7B model to encode label descriptions into 4096-dimensional text features. These text features, augmented with dataset embedding of the same dimensionality, are then employed as node features input into the GraphSAGE. When forming a minibatch from multiple datasets, we evenly sample 3 images per dataset within a batch for each GPU. For all images, We first apply random resizing with a ratio ranging from 0.5 to 2, followed by a random crop operation to achieve a final image size of 768 × 768 pixels. We use AdamW optimizer [27] with warmup and polynomial learning rate decay, starting with a learning rate of 0.0001. We train our model for 300k iterations on four 80G A100 GPUs.

### 4.1 Comparison on Multiple Datasets

In Table 2, we present the accuracy of our methods and compare them to other approaches on the seven training datasets. We use mean Intersection over Union (mIoU) to quantify the performance of models, a common metric used to evaluate the performance of segmentation models. Different methods adopt various approaches to construct their label spaces: *Dataset-Specific* represents a lack of a unified label space, where the model outputs a separate label space for each dataset. *Manually*

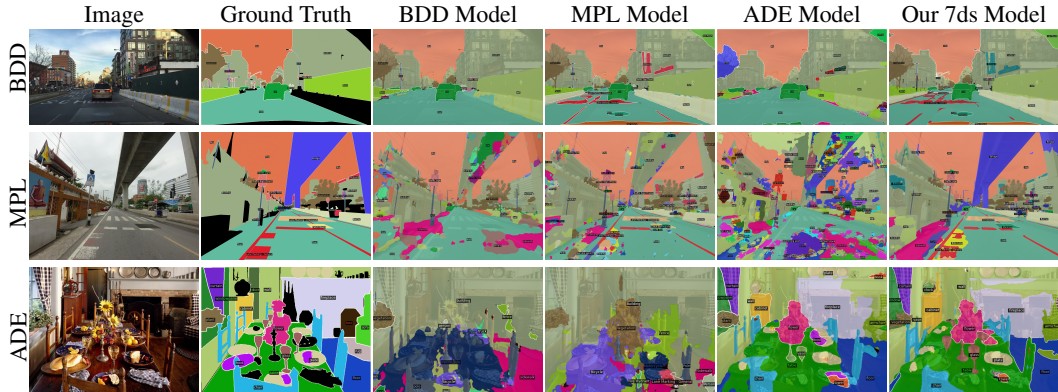

Figure 3: Visual comparisons with *Single dataset* model on different training datasets.

*Relabel* means manually re-annotating each image. *Manually Construct* means the label space is constructed through human expertise. *Automatically Construct* includes methods where the unified label space is automatically constructed by the model. We also establish two baseline methods: *Single dataset* and *Multi-SegHead*. *Single dataset* demonstrates the results of training on individual dataset only, while *Multi-SegHead* trains on multiple datasets by using dataset-specific segmentation heads.

The results demonstrate that our method achieves the best average performance in multi-dataset training, while also achieving significant performance improvements on datasets with a large number of classes such as the ADE and COCO datasets. We attribute this to the construction of a robust unified label space. Leveraging visual connections from the samples, our unified label space can discover label relationships beyond textual similarities. For instance, the visual appearance of the *fireplace* in ADE is similar to the *tunnel* in Mapillary. Despite their different semantic meanings, our model merges these labels for prediction, as detailed in subsection 4.5. This approach saves model capacity and facilitates knowledge transfer across datasets for improved prediction.

In Table 3, as a supplement to Table 2, we compare our model with various models trained on single dataset, as well as each segmentation head output of the *Multi-SegHead*. Detailed data for each dataset can be found in the Appendix D. From the table, it can be observed that training with multiple datasets helps improve the model's generalization performance. However, the performance of different segmentation head outputs in *Multi-SegHead* model shows significant differences due to the lack of a unified form of output that performs well across all datasets. In contrast, our approach provides a unified label space covering all datasets, resulting in a significant advantage in average performance. We also list the performance on the five unseen test datasets mentioned in Table 1. To evaluate the performance of our model on unseen datasets, we first evaluate the model results on its training dataset. We search for the optimal label mappings based on the accuracy of label predictions. There are no updates to any model parameters except the label mappings in this process. This process can actually be done manually without any annotation information. The results indicate that our model exhibits better generalization performance. It can handle various scenarios and consistently achieve excellent performance on the test set. Compared to *Multi-SegHead model*, our automatically constructed label space has advantages over dataset label spaces. The unified label space can integrate the semantic information from multiple dataset label spaces.

Figure 3 presents the segmentation results on multiple datasets predicted in unified label space. Compared to models trained only on *Single dataset*, our model successfully provides consistent predictions on all datasets. It's worth noting that in the BDD dataset, annotations are not provided for *lane marking*, *crosswalk* and *manhole*, which are only annotated in the Mapillary dataset. Our model successfully integrates the label space of the Mapillary dataset, thereby predicting these classes in the BDD dataset. More results are presented in Appendix F.

## 4.2 Results on WildDash 2 Benchmark.

WildDash 2 [47] provides a benchmark for semantic segmentation, designed to test the robustness of algorithms in real-world driving scenarios. Due to the insufficient number of training samples

Table 4: Performance comparison on WildDash 2 benchmark.

| Model | Venue | Trained | Meta Avg mIoU Class | Classic mIoU Class | Classic iIoU Class | Classic mIoU Cat. | Classic iIoU Cat. | Negative mIoU class |
|-------|-------|---------|---------------------|--------------------|--------------------|-------------------|-------------------|---------------------|
| EffPS [31] | IJCV 21 | ✗ | 32.2 | 35.7 | 24.4 | 63.8 | 56.0 | 20.4 |
| MSeg [23] | CVPR 20 | ✗ | 35.2 | 38.7 | 35.4 | 65.1 | 50.7 | 24.7 |
| SeamSeg [34] | CVPR 19 | ✗ | 37.9 | 41.2 | **37.2** | 63.1 | **58.1** | 30.5 |
| UniSeg [19] | ECCV 22 | ✗ | 39.4 | **41.7** | 35.3 | **65.8** | 57.4 | 34.8 |
| Ours | - | ✗ | **40.5** | 41.2 | 33.7 | 65.4 | 54.2 | **43.4** |
| SNpRN152 [3] | arXiv 20 | ✓ | 45.4 | 48.9 | 42.7 | 70.1 | 64.8 | 32.5 |
| NLL+ [4] | WACV 22 | ✓ | 46.8 | 51.0 | 43.9 | 71.4 | 65.5 | 32.6 |
| Uni NLL+ [5] | IJCV 24 | ✓ | 46.9 | 51.6 | 45.9 | 72.8 | 67.5 | 29.0 |
| FAN [45] | arXiv 22 | ✓ | 47.5 | 50.8 | 44.0 | **74.2** | 67.5 | 34.4 |
| MIX6D [26] | arXiv 22 | ✓ | 48.5 | 51.2 | 46.5 | 72.4 | 66.1 | 40.8 |
| Ours | - | ✓ | **50.0** | **52.2** | **47.5** | 72.4 | **68.6** | **44.6** |

Table 5: Comparison of Different Methods of Construct Label Spaces.

| Methods | $|L|$ | CS | MPL | SUN | BDD | IDD | ADE | COCO | Mean |
|---------|-------|-----|-----|-----|-----|-----|-----|------|------|
| Direct concatenation | 448 | 79.2 | 38.8 | 46.9 | 64.3 | 66.7 | 33.3 | 38.2 | 52.5 |
| Clustered by text features | 329 | 79.9 | 40.8 | 47.3 | 65.6 | 68.6 | 33.8 | 38.0 | 53.4 |
| Without GNN training | 231 | 79.8 | 39.7 | **47.7** | **66.1** | 68.1 | 37.8 | 45.3 | 54.9 |
| Without GPT's label description | 226 | 80.2 | 43.4 | 47.2 | 64.7 | **68.8** | 40.1 | 46.0 | 55.8 |
| Our proposed method | 217 | **80.7** | **43.7** | 47.5 | 65.5 | 68.6 | **42.0** | **46.7** | **56.4** |

provided by this dataset, the official recommendation is to use multiple datasets for training. Therefore, this benchmark is well-suited to evaluate the effectiveness of multi-dataset training methods. The WildDash 2 dataset includes negative test cases to challenge the robustness of the model. These negative test cases mainly consist of unconventional driving scenarios, and even non-driving scenarios. Across all pixels within negative test images, a robust model is expected to predict the *void* label for open-set classes and anomalous objects. The WildDash 2 benchmark refers to the metric named Meta Avg mIoU Class, which calculates the mean Intersection over Union for each class by weighting negative and positive test cases according to their occurrence in the benchmark dataset.

Table 4 presents the current results on the WildDash 2 leaderboard. We present results for zero-shot generalization using our 7ds model. To evaluate in an unseen setting, we map the non-evaluated classes in the unified label space to a *void* label for both positive and negative test frames. To ensure a fair comparison with other works, we also provide evaluation results for models trained using the training datasets from the Robust Vision Challenge 2022[1], which include CityScapes, ADE20K, Vistas, VIPER [36], ScanNet, and WildDash 2. The results indicate that our method achieves state-of-the-art performance in both zero-shot and trained settings. Our method exhibits significant performance improvements compared to other methods on negative test cases. We attribute this to the robustness of our model, which has been trained on diverse datasets, enabling it to perform well in unconventional scenarios. Our zero-shot model has been trained on a wider range of datasets compared to the trained model. Therefore, even without training on the WildDash 2 dataset, it achieves similar performance on negative test cases.

### 4.3 Ablation Study

To further explore the ability of our GNNs to construct a unified label space, we compared it with four alternative methods. The first method concatenates all dataset label spaces into a single unified space $\bigcup_{i=1}^{D} L_i$. The second approach constructs a unified label space by clustering text features based on cosine similarity using the DBSCAN [7] algorithm. In the third method, we train a segmentation network using an initial adjacency matrix, as outlined in Appendix B, without incorporating GNN training. The final method involves an ablation experiment, removing the label description module to observe its impact. Experimental results, presented in Table 5, demonstrate that the label space constructed based on GNNs can better assist in the learning of segmentation models. Unlike the first approach, our method optimizes model capacity by focusing on label relationships rather than dataset recognition. Compared to the second approach, our method can differentiate between classes with identical names but differing levels of granularity. By leveraging label descriptions to enrich semantic context, our approach constructs a more refined and functional label space.

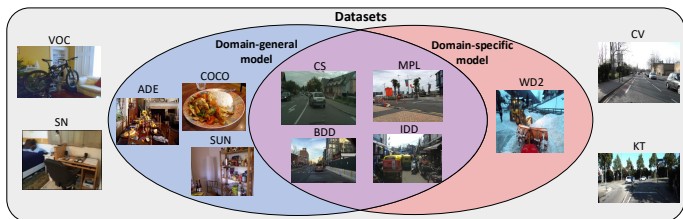

Figure 4: The composition of the training datasets.

Table 6: Performance on unseen dataset.

| Model | Driving | | Non-Driving | |
|---|---|---|---|---|
| | CV | KT | VOC | SN |
| General | 71.1 | 65.5 | **69.2** | **40.6** |
| Specific | **74.4** | **67.4** | 28.7 | 8.1 |

## 4.4 Exploring the Impact of Training Datasets

To explore the impact of training dataset selection on model performance, we train a domain-specific model focusing on road driving scenes and a domain-general model on more datasets, as shown in Figure 4. We conduct four sets of comparisons to evaluate the performance across datasets that were trained on both models, trained on one model and not on the other, and not trained on either model. As shown in Table 6 and Appendix E, the domain-specific model can focus more on learning features specific to the particular scene, resulting in slightly better performance on both trained and unseen driving scene datasets compared to domain-general model. On the other hand, domain-general model trained on more scenes and more data exhibit better generalization performance. Therefore, while it does not lag too far behind in performance on driving scene datasets, it demonstrates overwhelming advantages on other scene datasets.

## 4.5 Qualitative Results

Figure 5 presents the qualitative analysis results of the label space learned by our model. We compare the label space learned by our model with the label space constructed using text features. The class *curb* in the IDD dataset actually encompasses both the classes *curb* and *barrier* in Mapillary, whereas the constructed label space by text features cannot handle such subclass/superclass relations. In contrast, our GNNs learned label space splits the IDD *curb* into two classes for prediction, effectively handling such label relationships. Similar situations also include the class *tunnel or bridge* in the IDD dataset. However, since the proportion of bridge pixels is orders of magnitude greater than that of tunnels ($10^8$ pixels for *bridge* and $10^4$ pixels for *tunnel*), a more reasonable approach is to merge it with the class *bridge*, as our GNNs have done. Additionally, it is worth noting that due to visual similarities, the Mapillary *tunnel* has been merged with the ADE *fireplace*. This does not actually introduce any conflict because no dataset simultaneously annotates both the *tunnel* and *fireplace*. The model will construct the label space in a way that facilitates its learning process.

## 5 Conclusion

We propose a novel approach that leverages graph neural networks to construct a unified label space for training semantic segmentation models across multiple datasets. Our method addresses the challenge of label conflicts in multi-dataset semantic segmentation and demonstrates performance improvements across various datasets. The unified label space generated during training, generalizes well to unseen datasets, showcasing the effectiveness of our approach.

**Broader Impact.** Our work explores the use of graph neural networks to unify label spaces between datasets, providing a new direction for achieving robust and efficient multi-dataset training. By enabling semantic segmentation models to be trained on multiple datasets with a unified label space, our method can potentially reduce human effort required for re-labeling images and facilitate the expansion of training datasets. This can lead to the development of models that are more universally applicable across various datasets, benefiting a wide range of applications.

**Limitations.** Although our approach does not require manual relabeling efforts, it still relies on fully annotated datasets for training, in contrast to weakly-supervised and unsupervised methods. We aim to explore ways to integrate these alternative methods in future research. Errors in the fully automated construction of a unified label space do present some safety risks for autonomous driving tasks. Therefore, we recommend introducing a manual review mechanism to address these issues. Additionally, using ChatGPT may generate inaccurate label descriptions, which could affect the

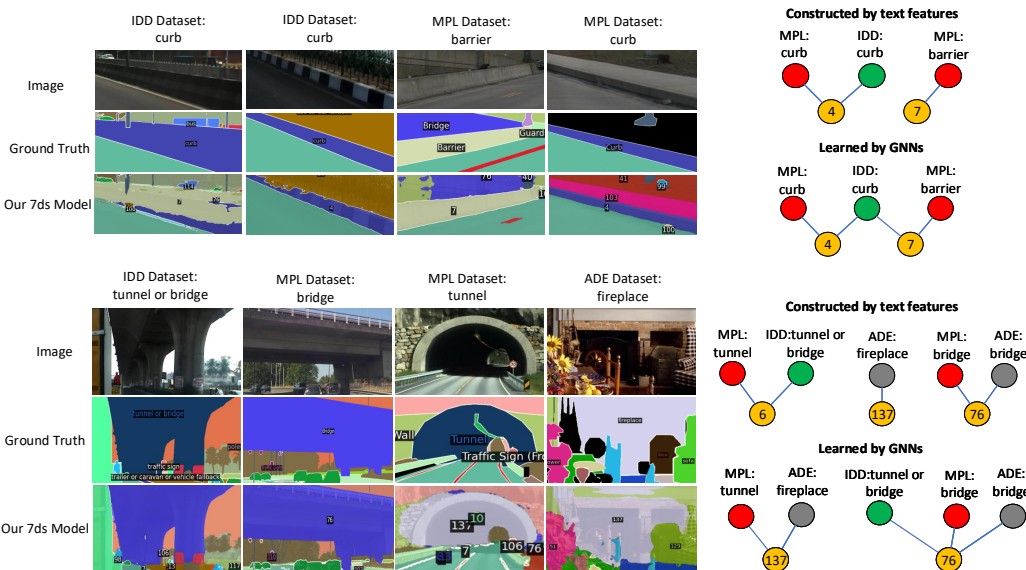

Figure 5: Comparison of unified label space learned by GNNs with constructed by text features.

prediction of label relationships. Therefore, we aim to improve the accuracy of label descriptions by incorporating label descriptions provided by official datasets as prompts.

# 6 Acknowledgments

This work was supported in part by NSFC Project (62176061), Shanghai Municipal Science and Technology Major Project (2021SHZDZX0103). The computations in this research were performed using the CFFF platform of Fudan University.

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

# A   Training Strategy

The training process is divided into three distinct stages: Multi-SegHead Training Stage, Alternating GNNs Training Stage and Alternating SegNet Training Stage, as shown in Algorithm 1. Initially, we commence with the Multi-SegHead Training Stage for 100k iterations. After completing Multi-SegHead Training Stage, we discard the multi-heads and initialize the UniSegHead for predicting class probabilities during subsequent training phases. Subsequently, we alternate between the Alternating GNNs Training Stage and Alternating SegNet Training Stage for a total of three cycles, each lasting 20k iterations. Notably, we prioritize the Alternating GNNs Training stage at the beginning of each cycle.

**Multi-SegHead Training Stage** aims to equip the segmentation network with fundamental segmentation capabilities. By commencing with a Multi-SegHead model, we provide a robust starting point for the subsequent stages, enhancing both performance and training efficiency. During this stage, the network is trained on multiple datasets using specific segmentation heads for each dataset. Each segmentation head comprises feature weights $\mathbf{W}_i$ of dimension $C \times |L_i|$, where $|L_i|$ represents the number of classes of dataset $i$. Pixel embedding $\mathcal{P}$ is multiplied by the $i$-th feature weight to make predictions within the $i$-th dataset's label space. This facilitates the utilization of dataset-specific annotations and cross-entropy loss for network training.

$$\mathcal{L}_m(\mathbf{y}, \mathbf{s}) = -\sum_{c=1}^{|L_i|} y_c \log(\text{softmax}(\mathbf{W}_i \mathbf{p})_c). \tag{9}$$

**Alternating GNNs Training Stage.** Training SegNet and GNNs simultaneously is challenging and impractical because training the segmentation network requires a discrete label mapping, while training the GNNs necessitates a continuous and differentiable adjacency matrix. Hence, we alternately training the GNNs and the segmentation network. The Alternating GNNs Training Stage focuses on optimizing a unified label space. During this stage, the GNNs are trained while keeping the segmentation network frozen. When mapping the prediction from the unified label space to the dataset label space, we directly multiply the adjacency matrix $\mathbf{M}_a$. We calculate the loss according to Equation 8.

**Alternating SegNet Training Stage.** During this stage, the GNNs remain frozen while the segmentation network is trained. We will use the technique detailed in Appendix C to convert the continuous adjacency matrix into a discrete label mappings that satisfies the constraints, and then use it to compute the cross-entropy loss function and train according to Equation 6. During the last Alternating SegNet training stage, we extend the training procedure to 100k iterations and drop the entire GNNs but retain the dataset label mappings $\{\mathbf{M}_i, i = 1, ..., K\}$. The unified label embedding $\mathbf{X}_u$ in this stage is no longer keep frozen and trained with the SegNet. In the latter part of this stage, we will use training dataset to evaluate the model's performance. Those links between the unified label space and dataset-specific label space which is not activated during the evaluation will be removed. After removing the unused connections, we continue training with the new label mappings. And thus, we obtain the final model. When applying the final model to predict in the wild, we could manually assign each unified label node a class name accord to their prediction and semantics.

# B   Selection of Unified Label Node Quantity and Initialization of Adjacency Matrix

The appropriate quantity of unified label nodes is crucial for balancing model performance and computational expenses. An excessive number of unified label nodes not only increases computational costs but may also result in redundancy in both the model and semantic space, leading to situations where multiple unified label nodes represent similar semantic concepts. Conversely, an insufficient number of unified label nodes can prevent the model from fully expressing the entire semantic space $\bigcup_{i=1}^{D} L_i$, making it difficult to learn certain classes in the datasets, thus impacting model performance. Inspired by [53], we determine the number of unified label nodes using the following steps. We first enumerate all feasible schemes for merging dataset labels, which form the Cartesian product of all dataset label sets $\mathbb{L} = L_1 \times L_2 \times \ldots \times L_K$. Label merging does not require the participation of all datasets simultaneously, so $L_i$ can be an empty element, indicating no participation in label merging.

**Algorithm 1** The training pipeline of our model

---

**Input:** the number of datasets $K$, the Segmentation Network $SegNet$, the Graph Neural Networks $GNNs$, Multi-head iters $I_p$, GNNs training iters $I_g$ and SegNet training iters $I_s$
Stage = Multi-head stage; iter = 0
**for** a multi-datasets sampled mini-batch $\{x_i, y_i\}_{i=1}^K$ **do**
    **if** Stage == Multi-SegHead Training:
        Calculate $\mathcal{L}_m$ by Equation 9
        Update $SegNet$ to minimize $\mathcal{L}_m$
        **if** iter++ > $I_p$: Replace Multi-SegHead with UniSegHead; Stage = GNNs training; iter = 0
    **if** Stage == GNNs training:
        Calculate $\mathcal{L}_g$ by Equation 8
        Update $GNNs$ to minimize $\mathcal{L}_g$
        **if** iter++ > $I_g$: Solve dataset label mappings by Algorithm 2; Stage = Seg training; iter = 0
    **if** Stage == SegNet training:
        Calculate $\mathcal{L}_{ce}$ by Equation 6
        Update $SegNet$ to minimize $\mathcal{L}_{ce}$
        **if** iter++ > $I_s$: Stage = GNNs training; iter = 0
    **end switch**
**end for**

---

We evaluate the quality of label merging using the predefined loss function $\mathcal{L}_c$, where merging classes with the same semantic meaning results in smaller losses. Additionally, we require that the merging satisfies the constraint: each dataset node has only one corresponding merged node. It is worth noting that this constraint is stronger than our label mapping constraint described in subsection 3.1, thus ensuring that our adjacency matrix is always valid. We formulate optimization objective function as:

$$\text{minimize}_x \quad \sum_{\boldsymbol{t} \in \mathbb{T}} x_{\boldsymbol{t}} E_{\mathcal{S}_k} \left[ \sum_{c \in L_k | \boldsymbol{t}(c)=1} \mathcal{L}_c \left( S_c^k, \tilde{S}_c^k \right) \right] + \lambda |L|$$
$$\text{subject to} \quad \sum_{\boldsymbol{t} \in \mathbb{T} | \boldsymbol{t}_c = 1} x_{\boldsymbol{t}} = 1 \quad \forall_c, \tag{10}$$

where $\mathbb{T}$ represents the set of all feasible edges connecting dataset label nodes to unified label nodes, $x_{\boldsymbol{t}} \in \{0, 1\}$ indicates whether edge $\boldsymbol{t} \in \mathbb{T}$ is selected, with $x_{\boldsymbol{t}} = 1$ indicating the presence of edge $\boldsymbol{t}$ and 0 indicating its absence. $\mathcal{S}_k$ denotes the dataset to which edge $\boldsymbol{t}$ belongs, $L_k$ represents the label space of $\mathcal{S}_k$, and $\boldsymbol{t}(c)$ is the label connected by edge $\boldsymbol{t}$. $\lambda |L|$ is the penalty term for the number of nodes, encouraging the optimizer to merge similar nodes to obtain a more compact unified label space. Based on experimental insights and referencing paper [52], we selected the hyperparameter $\lambda = 0.5$. $\mathcal{L}_c \left( S_c^k, \tilde{S}_c^k \right)$ is primarily used to evaluate the quality of label merging, where $S_c^k$ represents the output results on class $c$ using the original dataset segmentation head, and $\tilde{S}_c^k$ represents the output results on class $c$ using segmentation heads from other datasets participating in the merging.

$$\mathcal{L}_c \left( S_c^k, \tilde{S}_c^k \right) = \sum_{s_c \in \tilde{S}_c^k} IoU(S_c^k) - IoU(s_c), \tag{11}$$

where, $IoU$ is a metric commonly used to measure the performance of semantic segmentation models. Therefore, we use this metric here to evaluate the assistance of merged nodes to the model. By using a linear optimization solver to solve the optimization objective Equation 10, we can obtain a unified label space. We use the number of labels in this label space as the quantity of our unified label nodes and initialize our adjacency matrix based on their label mapping relationships.

## C   Solving the dataset Label Mappings

Given the definition of the label mappings and the constraints outlined in subsection 3.1, solving the dataset label mappings could be conceptualized as a weighted bipartite graph matching problem. The weight of each edge is determined by the values of the GNNs learnable adjacency matrix. The objective is to maximize the sum of edge weights while ensuring that each node of the specific dataset is connected. The unbalanced optimal transport algorithm offers an approximate solution to such

---
**Algorithm 2** The procedure of solving the dataset label mappings
---

**Input:** Submatrix of adjacency matrix for each dataset $\{S^{(i)}, i = 1, ..., K\}$, the classes of each
dataset $\{c^{(i)}, i = 1, ..., K\}$, the number of unified label node $N$
**Output:** Mapping matrices $M^{(1)}, ..., M^{(K)}$

Initialize $\alpha = \frac{1}{\mathbf{N}}\mathbf{1}_{\mathbf{N} \times 1}$, $\beta^{(i)} = \{\frac{1}{c^{(i)}}\mathbf{1}_{c^{(i)} \times 1}, i = 1, ..., K\}$, $d = 0$, $\mu$
**for** $i = 1$ to $K$ **do**

$\quad Q^{(i)}_{c^{(i)} \times \mathbf{N}} = \text{UOT}(\alpha, \beta^{(i)}, S^{(i)})$

$\quad$ # Assign the label to the node with the highest score

$\quad M^{(i)}_{j,k} = \arg\max_j \{Q^{(i)}_{j,1}, ..., Q^{(i)}_{j,\mathbf{N}}\}$

$\quad$ **for** $j = 1$ to $c^{(i)}$ **do**

$\quad\quad$ # Find an unlinked dataset-specific node

$\quad\quad$ **if** $\max M^{(i)}_j == 0$ **then**

$\quad\quad\quad q_j = \text{sort}(Q^{(i)}_j)$

$\quad\quad\quad$ # Find first multi-mapped node, where $sum(M^{(i)}_{j'}) > 1$

$\quad\quad\quad j', k' = \text{findFirstMulMapped}(q_j, M^{(i)})$

$\quad\quad\quad$ # Replace the corresponding label mappings

$\quad\quad\quad M^{(i)}_{j',k'} = 0, M^{(i)}_{j,k'} = 1$

$\quad\quad$ **end if**

$\quad$ **end for**

$\quad \bar{\beta}^{(i)}_j = \sum_{k=1}^{c^{(i)}} Q^{(i)}_{j,k}$

$\quad \beta^{(i)} = \mu\beta^{(i)} + (1 - \mu)\bar{\beta}^{(i)}$

**end for**

---

problems, but it does not guarantee adherence to the constraint of connecting each node in the specific dataset. Therefore, based on the optimal matching result obtained from the algorithm, we employ a greedy strategy to adjust the matching scheme for all unconnected nodes within the specific dataset.

Initially, for every unconnected dataset-specific node, we sort all unified label nodes based on their matching preferences, from highest to lowest. Subsequently, we examine these unified label nodes and their associated dataset-specific nodes. In cases where the associated dataset-specific node is also connected by another unified label node, we adjust the edge so that this unified label node links to the unconnected specific dataset node, following this process until all nodes are connected. The specific procedure is outlined in Algorithm 2.

## D    Comparison with Baselines on Multiple Datasets

Table 7 lists the results of mutual evaluation among training datasets. The results indicate that individually-trained models generally demonstrate good accuracy when tested on the same dataset, but perform poorly on other datasets. Our method can leverage knowledge from multiple datasets to improve performance on individual datasets. Table 8 lists the results of the Multi-SegHead model's predictions on different datasets using different segmentation head outputs. It can be observed that none of the segmentation head outputs could consistently provide accurate predictions across all datasets, indicating that their label spaces do not effectively cover all datasets. Table 9 and Table 10 list the performance of different models on unseen datasets. The results indicate that our method has better generalization performance on unseen datasets compared to other methods. Additionally, the unified label space we construct contains richer semantic information, enabling flexible adaptation to datasets from different scenarios.

Table 7: Semantic segmentation accuracy (mIoU) on training datasets compared with Single dataset model.

| Train\Test | CS | MPL | SUN | BDD | IDD | ADE | COCO | Mean |
|---|---|---|---|---|---|---|---|---|
| CS | 77.0 | 7.2 | 3.7 | 43.4 | 30.4 | 2.6 | 2.8 | 23.9 |
| MPL | 65.6 | 30.2 | 4.8 | 54.5 | 48.4 | 4.7 | 5.9 | 30.6 |
| SUN | 13.0 | 3.4 | 43.9 | 12.4 | 9.5 | 5.1 | 3.0 | 12.9 |
| BDD | 58.9 | 12.1 | 3.6 | 62.4 | 30.9 | 2.7 | 2.7 | 24.8 |
| IDD | 50.9 | 10.4 | 4.3 | 43.5 | 66.8 | 3.1 | 2.8 | 26.0 |
| ADE | 42.7 | 11.5 | 35.5 | 34.7 | 27.5 | 34.5 | 12.8 | 28.5 |
| COCO | 45.5 | 14.0 | 29.1 | 42.0 | 30.1 | 14.8 | 38.0 | 30.5 |
| Ours | **80.7** | **43.7** | **47.5** | **65.5** | **68.6** | **42.0** | **46.7** | **56.4** |

Table 8: Semantic segmentation accuracy (mIoU) on training datasets compared with Multi-SegHead.

| Label space\Test | CS | MPL | SUN | BDD | IDD | ADE | COCO | Mean |
|---|---|---|---|---|---|---|---|---|
| CS | 79.5 | 16.1 | 10.4 | 61.6 | 42.8 | 2.5 | 3.6 | 30.9 |
| MPL | 72.9 | 36.1 | 12.9 | 60.4 | 54.7 | 7.0 | 8.9 | 36.1 |
| SUN | 33.5 | 9.3 | 47.3 | 29.5 | 22.7 | 6.4 | 5.3 | 22.0 |
| BDD | 67.1 | 16.6 | 8.9 | **65.6** | 39.0 | 2.6 | 3.6 | 29.1 |
| IDD | 72.3 | 20.3 | 10.8 | 59.8 | 67.0 | 3.3 | 4.3 | 34.0 |
| ADE | 62.7 | 20.8 | 43.5 | 57.6 | 46.6 | 30.5 | 20.5 | 40.3 |
| COCO | 59.5 | 21.4 | 34.4 | 57.5 | 44.4 | 18.0 | 36.7 | 38.8 |
| Ours | **80.7** | **43.7** | **47.5** | 65.5 | **68.6** | **42.0** | **46.7** | **56.4** |

Table 9: Semantic segmentation accuracy (mIoU) on unseen datasets compared with Single dataset.

| Train\Test | KT | SN | CV | VOC | CT | Mean |
|---|---|---|---|---|---|---|
| CS | 62.3 | 4.3 | 67.5 | 16.1 | 7.0 | 31.4 |
| MPL | **67.3** | 6.1 | **73.1** | 24.6 | 13.3 | 36.9 |
| SUN | 12.4 | 32.2 | 20.2 | 13.2 | 5.4 | 16.7 |
| BDD | 56.0 | 5.3 | 66.7 | 17.3 | 6.7 | 30.4 |
| IDD | 50.6 | 6.1 | 62.6 | 17.7 | 7.4 | 28.9 |
| ADE | 45.6 | 25.8 | 51.7 | 39.0 | 23.7 | 37.2 |
| COCO | 45.4 | 27.4 | 53.9 | 63.0 | 35.8 | 45.1 |
| Ours | 65.5 | **40.6** | 71.1 | **69.2** | **38.1** | **56.9** |

Table 10: Semantic segmentation accuracy (mIoU) on unseen datasets compared with Multi-SegHead.

| Label space\Test | KT | SN | CV | VOC | CT | Mean |
|---|---|---|---|---|---|---|
| CS | 59.7 | 14.6 | 72.3 | 29.0 | 9.3 | 37.9 |
| MPL | 63.7 | 18.3 | **72.8** | 33.6 | 16.4 | 41.0 |
| SUN | 27.0 | **42.9** | 34.9 | 29.9 | 10.0 | 28.9 |
| BDD | 61.0 | 15.7 | 71.2 | 21.9 | 8.7 | 35.7 |
| IDD | 56.8 | 16.3 | 69.0 | 29.8 | 10.3 | 36.4 |
| ADE | 56.4 | 40.8 | 65.1 | 56.2 | 30.7 | 49.8 |
| COCO | 55.9 | 37.6 | 64.6 | **69.6** | **38.2** | 53.2 |
| Ours | **65.5** | 40.6 | 71.1 | 69.2 | 38.1 | **56.9** |

Table 11: Performance on both trained datasets.

| Model | Trained | Driving scene dataset | | | |
|---|---|---|---|---|---|
| | | CS | MPL | BDD | IDD |
| General | ✓ | 80.7 | 43.7 | 65.5 | 68.6 |
| Specific | ✓ | **82.2** | **45.7** | **68.8** | **71.4** |

Table 12: Unseen domain-general model vs. Trained domain-specific model.

| Model | Trained | Driving scene dataset |
|---|---|---|
| | | WD2 |
| General | ✗ | 40.5 |
| Specific | ✓ | **50.2** |

Table 13: Trained domain-general model vs. Unseen domain-specific model.

| Model | Trained | Non-Driving scene dataset | | |
|---|---|---|---|---|
| | | SUN | ADE | COCO |
| General | ✓ | **47.5** | **42.0** | **46.7** |
| Specific | ✗ | 4.4 | 5.4 | 7.3 |

Table 14: Performance on both unseen datasets.

| Model | Trained | Driving | | Non-Driving | |
|---|---|---|---|---|---|
| | | CV | KT | VOC | SN |
| General | ✗ | 71.1 | 65.5 | **69.2** | **40.6** |
| Specific | ✗ | **74.4** | **67.4** | 28.7 | 8.1 |

# E   Exploring the Impact of Training Datasets

Tables 11 to 14 present a performance comparison between the domain-general and domain-specific models across various training settings and datasets. Figure 6 illustrates the segmentation results of these models on the different datasets.

**Performance Comparison on Both Trained Datasets.** From Table 11, it can be concluded that the performance of the domain-specific model focusing on driving scenes is higher on all trained datasets compared to the domain-general model, despite the domain-general model being trained on five times more samples than the domain-specific model. This is likely attributed to the fact that the domain-specific model needs to predict fewer classes and is less affected by label space conflicts. The model can therefore focus on learning features specific to the particular scene.

**Performance Comparison: Unseen Domain-general Model vs. Trained Domain-specific Model.** In Table 12, we selected a driving scene dataset that the domain-specific model was trained on, while the domain-specific model was not trained on it, for comparison. As expected, the domain-specific model demonstrates superior performance, and the domain-general model closely follows suit.

**Performance Comparison: Trained Domain-general Model vs. Unseen Domain-specific Model.** In Table 13, we selected several non-driving scene datasets that the domain-general model was trained on, while the domain-specific model was not trained on it. The domain-specific model fails to predict in non-driving scenes. Compared to Table 12, the generalization performance of the domain-specific model in unseen scenes markedly trails behind the domain-general model. Although the domain-specific model may have encountered similar objects in driving scene datasets like *wall*, it still struggles to predict these seen objects well in non-driving scene datasets, as shown in Figure 6.

**Performance Comparison on Both Unseen Datasets.** In Table 14, we selected datasets that were unseen to both models, including driving scene datasets and non-driving scene datasets. We can observe that the generalization performance of the domain-specific model is slightly better in driving scenes, but its performance in other scenes lags far behind that of the domain-general model. Overall, models trained on more scenes and more data tend to achieve better generalization performance.

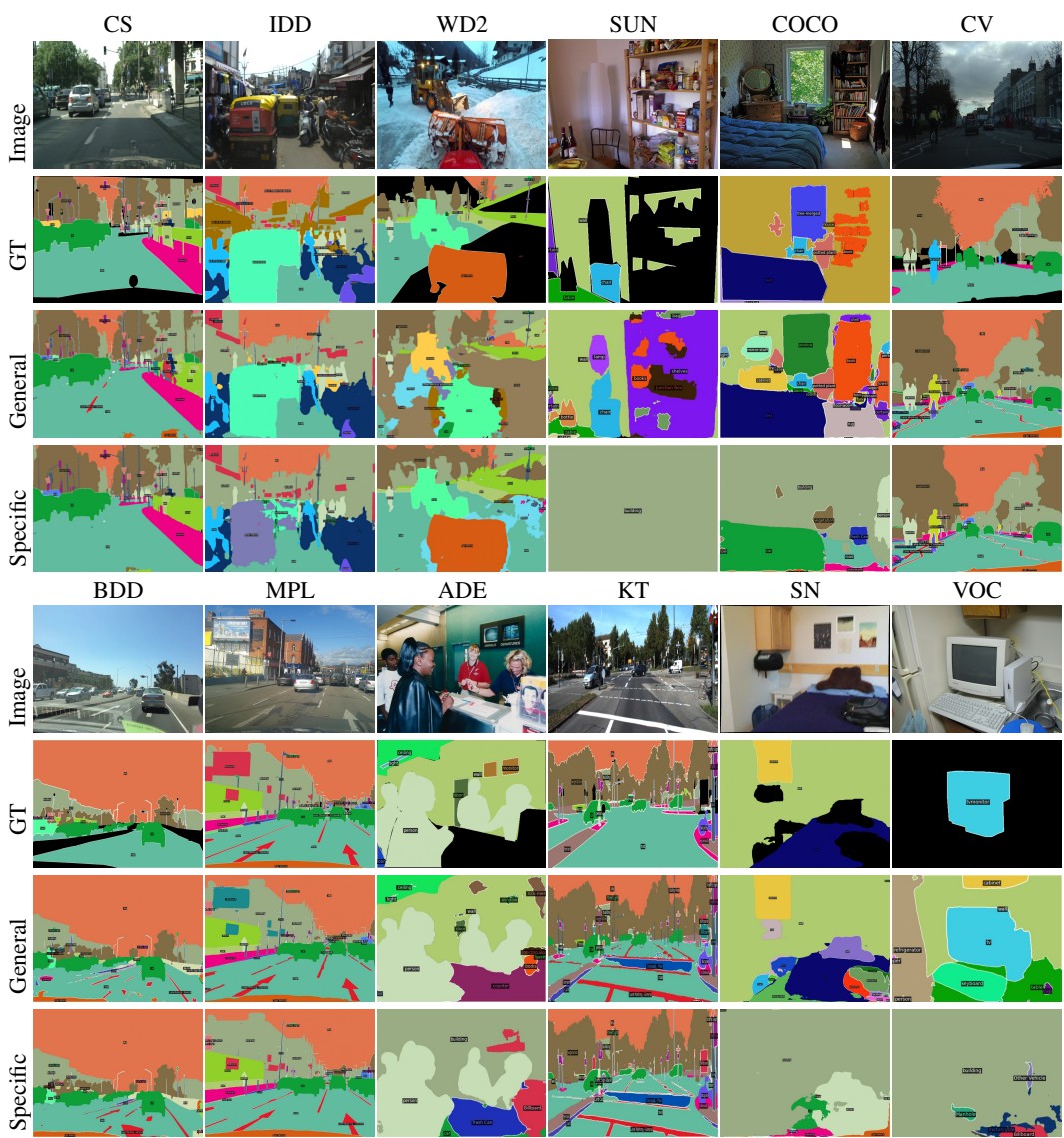

Figure 6: Visual comparisons of different training dataset models.

# F Visualization

Figure 7 presents the visual comparisons of models trained on single dataset and our model predicted in universal label space. From the figure, it is evident that our model achieves consistently strong performance across all training datasets while integrating label spaces from different datasets. For example, it predicts class *lane marking* and *crosswalk* for the ADE and BDD datasets, and predicts class *books* for the SUN dataset. Figure 8 shows our universal predictions on test datasets. The results demonstrate that our method generalizes well across multiple unseen datasets from different domains. Figure 9 shows the visual comparisons of different models on the WildDash 2 benchmark.

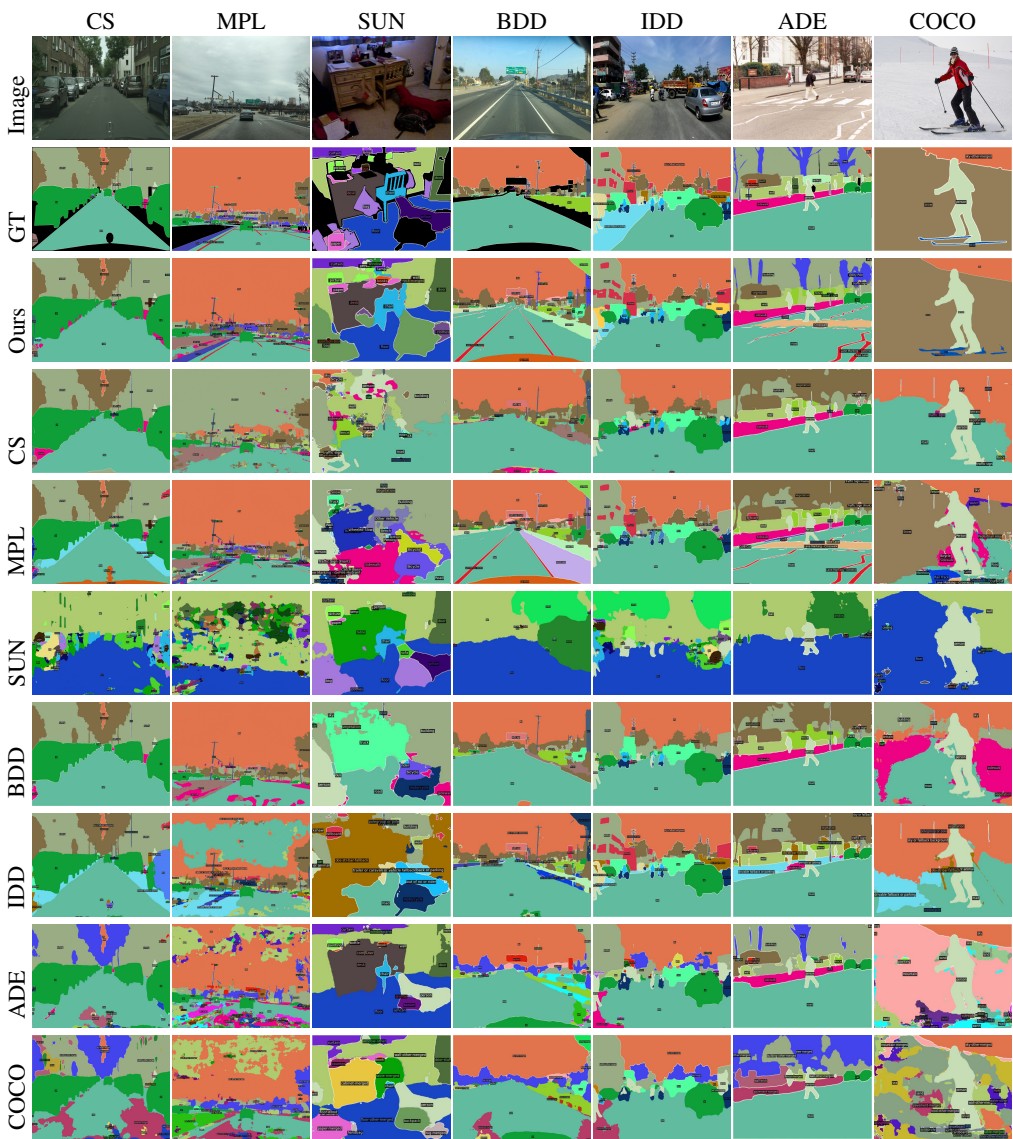

Figure 7: Visual comparisons on training datasets.

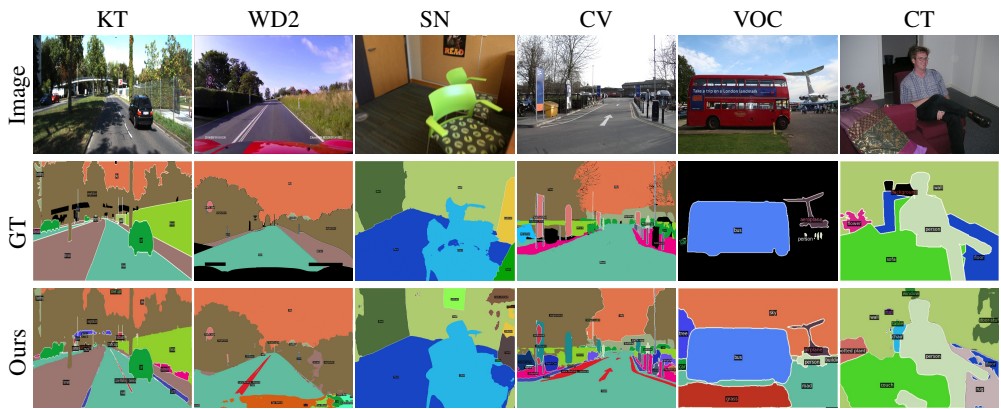

Figure 8: Visual comparisons on unseen test datasets.

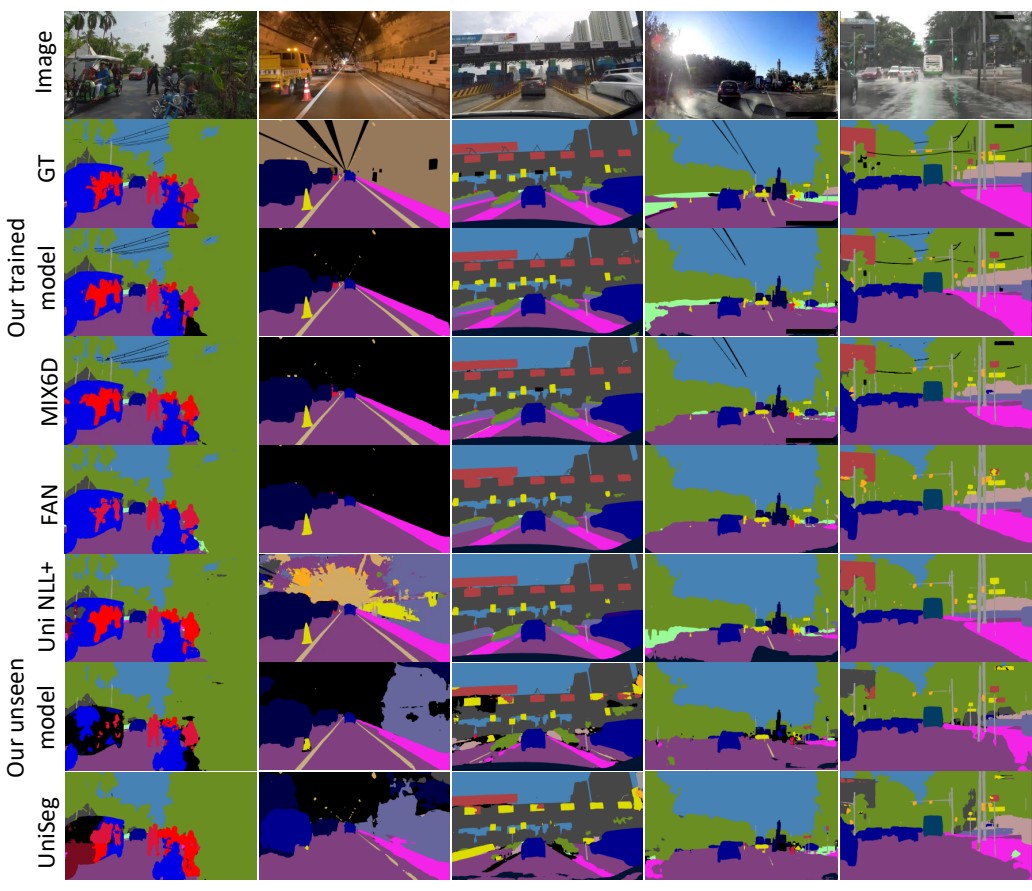

Figure 9: Visual comparisons on WildDash 2 benchmark.

