# OpenReview forum: "Automated Label Unification for Multi-Dataset Semantic Segmentation with GNNs"
_NeurIPS.cc/2024/Conference — NeurIPS 2024 poster_

### Official Review · Reviewer_pNQc · 2024-07-12

**Soundness:** 2
**Presentation:** 3
**Contribution:** 3
**Rating:** 4
**Confidence:** 4

**Summary:**

The paper proposes a multi-dataset training approach that works on top of a unified output taxonomy mapped onto individual dataset-specific taxonomies. First, a multi-head model is trained. Then, a unified taxonomy is automatically constructed by merging semantically identical classes, determined based on segmentation performance on individual classes when a connection is made. After that, GNNs are used to further refine the mapping from the individual dataset-specific classes to the unified taxonomy.

The input nodes of the GNN are dataset-specific labels and unified classes. Dataset-specific labels are first enriched with a textual description provided by ChatGPT, which is then encoded with LLaMA-2 and combined with a trainable dataset embedding to create node input features. The addition of the trainable dataset embedding enables the model to learn the different meanings of classes with the same name in different datasets.

Finally, the training process alternates between fine-tuning the segmentation network and refining the mappings with the GNN.

**Strengths:**

1. Well-written and clear.
2. Achieved state-of-the-art results on the WildDash 2 benchmark, demonstrating improvements over previous work.
3. Introduces a novel approach for label unification.

**Weaknesses:**

1. The paper primarily focuses on individual benchmark performance rather than the quality of the recovered taxonomy. In practical scenarios, it's crucial to reason within a unified label space. Mistakes such as equating MPL:tunnel to ADE:fireplace could mean the difference between advancing or halting in a real-world scenario.
2. The model achieving state-of-the-art performance on WildDash is trained exclusively on driving datasets, unlike other methods submitted that were trained on the full RVC collection (including ADE20K and COCO), potentially impacting performance in road driving contexts.
3. There is significant variation in training design choices in Table 2, making it challenging to isolate the impact of each individual choice (e.g., backbone, selection of dataset-specific classes, etc.).
4. The initialization of the adjacency matrix appears to be a crucial step for model performance, yet it is not discussed in the main paper.

**Questions:**

1. What is the size of the created taxonomy in different tables (e.g. in Table 5)? Furthermore, Table 5 could also include the performance after initializing the adjacency matrix, without additional GNN training.
2. How does the quality of the created taxonomy compare to the manual taxonomy provided by [5]? Are all relationships recovered? For example, IDD:tunnel is not connected to MPL:tunnel in Figure 5.
3. How are unseen datasets mapped to the unified taxonomy in Table 3? Is it done manually or automatically?
4. Is there any advantage to using the formulation in Eq. 6 as opposed to NLL+ [4]?
5. How much does the trained taxonomy differ from the initialized taxonomy? Do all unified classes "survive" after training, or do some become obsolete?
6. What exactly is the zero-shot model on WildDash 2 in Table 4?
7. How long does it take to initialize the adjacency matrix? What exactly is the difference compared to [52]? Are there similarities to [5], such as pairwise merging?

**Limitations:**

1. Focus on individual benchmark performance, rather than the process and the quality of the produced taxonomy.

---

> ### Author Rebuttal · Authors · 2024-08-07
>
> Thank you for your thorough review and insightful comments regarding our manuscript. Below are our responses to your questions and concerns.
>
> **W1: Focus on benchmark performance vs. taxonomy quality**
>
> We acknowledge the importance of reasoning within a unified label space. Our focus on multi-dataset training aims to automatically integrate labels across datasets. Evaluation on these datasets is a straightforward approach. To assess the quality of our unified label space, we conducted an indirect evaluation using WD2, demonstrating its effectiveness in unseen scenarios.
>
> **W2: Impact of training exclusively on driving datasets**
>
> We recognize the potential concern regarding dataset bias. It's worth noting that other methods on the WD2 leaderboard also use datasets beyond RVC. In the revised version, we will include a model trained on the full RVC collection for a more comprehensive comparison.
>
> **W3: Difficulty isolating individual training design choices**
>
> We apologize for the lack of clarity in Table 2. Due to long training times and limited access to open-source methods, comprehensive comparative experiments are challenging. Our primary comparison highlights the significant improvement our model offers over the single dataset and multi-SegHead baselines.
>
> **W4: Lack of discussion on adjacency matrix initialization**
>
> We regret this omission and have conducted ongoing experiments to address it. Results indicate that our proposed initialization method (Algorithm 2) provides a slight improvement over randomized initialization on a 3ds setting, suggesting a relatively minor impact on overall performance.
>
> | Method|\|L\||CS|SUN|CV|Mean|
> |--|--|--|--|--|--|
> |Randomized adjacency matrix|54| 78.0|43.1|82.4|67.8
> |Without GNN training|Pending|Pending| Pending|Pending|Pending
> |Ours| 54|78.4|43.3|82.6|68.1
>
> ---
>
> **Q1: Size of the created taxonomy in different tables and performance after initializing the adjacency matrix.**
>
> Thank you for your suggestion. We agree that including the initialization of the adjacency matrix in Table 5 would enhance reader understanding. However, due to time constraints, we cannot currently provide results for all seven datasets. These findings will be included in the revised version of the paper. Comparative experiments in a smaller setting were conducted, as addressed in our response to Weakness 4. Below is a summary of label space sizes for the methods, which we will incorporate into the revised version.
>
> | Variants | \|L\| |
> |--|--|
> |1 | 448 |
> |2 | 329 |
> |3 | 226 |
> |Ours | 217 |
>
>
> **Q2: Comparison of created taxonomy with the manual taxonomy.**
>
> To compare our constructed taxonomy with the manual taxonomy provided by [5], we adopted [5] as the ground truth standard, assessing whether categories from all datasets were appropriately linked. Categories were categorized into Merged, Single, and Split classes, each evaluated based on Correct, Partially Correct, and Wrong Classifications. We introduced *Wrong but reasonable* to accommodate justified connections that were nonetheless classified as wrong. For instance, while the Cityscapes dataset marked the Traffic sign (back) as ignored, [5] connected it based on expert knowledge.
>
> The summary of our 7ds learned taxonomy is as follows:
>
> | |Correct|Partial Correct|Wrong|Total Num|Wrong but reasonable |
> |--|--|--|--|--|--|
> |Merged|137|108|47|292|17 |
> |Single|38|12|16|66|6 |
> |Split|1|3|86|90|23 |
> |Total Num|176|123|149|448|46 |
>
> An initial taxonomy calculated using Algorithm 2 yields:
>
> | |Correct|Partial Correct|Wrong|Total Num|Wrong but reasonable |
> |--|--|--|--|--|--|
> |Merged|131|115|46|292|6 |
> |Single|44|7|15|66|3 |
> |Split|0|0|90|90|20 |
> |Total Num|175|122|151|448|29 |
>
> Notably, our constructed unified label space contains only 217 categories, representing a 6% reduction from the 231 categories formed by Algorithm 2’s initial adjacency matrix, while achieving nearly consistent quality in taxonomy recovery.
>
> **Q3: Mapping of unseen datasets to the unified taxonomy.**
>
> As indicated in Lines 212-214 of our paper, the mapping process is conducted automatically. We identify the optimal mapping by comparing the unified label categories predicted by our model on the training set of the unseen datasets against the ground truth labels.
>
> **Q4: Advantages of using the formulation in Eq. 6 compared to NLL+ [4].**
>
> We regret that because [4] is not open-sourced, our reproduced version of NLL+ does not converge under our training framework, prompting us to utilize Eq. 6 for training instead.
>
>
> **Q5: Differences between trained taxonomy and initialized taxonomy.**
>
> For the differences in the label space, please refer to Q2 and the global rebuttal. The initial unified label space had 231 categories, but some became obsolete during training, resulting in 217 categories in the final 7ds-trained model. Our paper (L487-480) explains the methods to eliminate inactive connections. Specifically, during the final training phase, we evaluate the model in the training dataset to remove inactive connections and unlinked nodes.
>
> **Q6: Explanation of the zero-shot model on WildDash 2.**
>
> The zero-shot model on WildDash 2, referenced in Table 4, is the model trained using our 7 datasets (i.e., the "ours" model in Table 2). We evaluated this model against the WildDash 2 dataset without any additional fine-tuning.
>
>
> **Q7: Time required to initialize the adjacency matrix and comparison with [52].**
>
> Using four 80G A100 GPUs, initializing the adjacency matrix takes approximately two days to train the Multi-SegHead model, followed by half a day of cross-evaluation across multiple datasets. Obtaining the initial adjacency matrix using Algorithm 2 takes around one hour. In comparison to [52], we modified the cost calculation to utilize the IoU metric. This approach is only applicable to category merging and shares similarities with the partial merge method in [5].
>
> We hope this clarifies your questions and appreciate your feedback.

---

> > ### Comment · Reviewer_pNQc · 2024-08-13
> > **Feedback on the rebuttal**
> >
> > I would like to thank the authors for the thorough feedback and additional experiments.
> >
> > Still, even after reading it, I am keeping my original score.
> >
> > The new results show that even though the proposed method improves the results on individual benchmarks it does so at the expense of relation discovery quality (Q2, W4, and corresponding answers by the authors). This is connected to my main concern (W1), and that is that individual benchmark performance is not a good proxy for evaluating the task of label unification.
> >
> > Furthermore, the WD2 performance is named as one of the strengths of this approach, but at this moment it is not clear if the improvements are due to the methodological contributions of this paper or the choice of training data. I am inclined to believe that it is due to omission of COCO and ADE20K classes in the final taxonomy. This is also suggested by the significantly worse performance of the zero-shot model which was trained on the Vistas dataset which should be enough for a good result on WD benchmark.
> >
> > With regards to W3, previous work comes to similar conclusions, so that is not enough of a contribution.

---

> ### Author Response · Authors · 2024-08-08
>
> We are pleased to inform you that we have obtained the complete experimental results regarding the initialization, answering W4 & Q1. Our experiments were conducted on three datasets (Cityscapes, Sunrgbd and CamVid) using two 32GB V100 GPUs. After completing 50,000 multi-SegHead training iterations, we performed 50,000 segmentation network training iterations for each group using the same multi-head model parameters. For the experiments involving GNN, an additional 30,000 GNN training iterations were included (without updating the segmentation network parameters). The results indicate that our GNN training shows a performance improvement compared to the results obtained using Algorithm 2 for initializing the adjacency matrix without GNN training. Algorithm 2 provides a good starting point for GNN training and contributes to the overall performance enhancement.
>
>
> | Method|\|L\||CS|SUN|CV|Mean|
> |--|--|--|--|--|--|
> |Randomized adjacency matrix with GNN training|54| 78.0|43.1|82.4|67.8|
> |Intialized adjacency matrix without GNN training|56|78.0| 42.0|82.6|67.5|
> |Ours (Intialized adjacency matrix with GNN training)| 54|78.4|43.3|82.6|68.1|

---

> ### Author Response · Authors · 2024-08-13
>
> Dear Reviewer,
>
> Thank you for your valuable feedback. We agree that evaluating a universal taxonomy instead of merely benchmark performance of downstream tasks is a crucial aspect of this research, particularly in terms of relation discovery quality. However, we currently face a challenge due to the lack of a dedicated benchmark specifically designed for assessing taxonomy quality. While the manually curated taxonomy provided by [5] serves as a helpful reference, it also incorporates expert knowledge that might not be well aligned with the visual features present in the dataset images. Furthermore, there is currently an absence of well-established metrics specifically for this type of evaluation. Given these current limitations, this work has primarily focused on evaluating model performance across different datasets. Moving forward, we are committed to exploring ways to better evaluate relation discovery quality and to address the concerns you have raised. We appreciate your insights and believe they are instrumental in guiding our future research endeavors.
>
> Regarding the WD2 performance, our zero-shot model achieved SOTA results compared to other zero-shot models on the leaderboard, which we believe highlights the contributions of our approach. It’s important to note that the Vistas dataset lacks annotations for the pickup, van, and autorickshaw categories present in WD2. Additionally, we did not utilize the relabeled data provided by WD2 for these categories in the MVD, Cityscapes, and IDD datasets during training. Therefore, given the dataset bias, it is expected that the zero-shot model would perform lower on WD2 compared to models specifically trained on the WD2 dataset.
>
> Thanks!

---

### Official Review · Reviewer_dye9 · 2024-07-17

**Soundness:** 3
**Presentation:** 3
**Contribution:** 2
**Rating:** 4
**Confidence:** 3

**Summary:**

This paper introduces a method using GNN to automatically construct a unified label space across multiple datasets, addressing the issue of label space conflicts in multi-dataset semantic segmentation. The method eliminates the need for manual re-annotation or iterative training, significantly enhancing the efficiency and effectiveness of model training. Experiments show that this method has certain effectiveness.

**Strengths:**

1. The motivation behind this approach is clear, and automatically constructing a unified label space across multiple datasets in multi-dataset semantic segmentation is meaningful.
2. The authors use numerous illustrations and provide a detailed description of the experimental parameters, making it easy to follow and reproduce.

**Weaknesses:**

1. This paper proposes a new method, but in my view, it is merely a collection of tricks. For example, $d_i$ is introduced in Equation (1), but the authors neither explain its role through ablation experiments nor from a methodological perspective. It is recommended that the authors provide a more detailed explanation of the motivation for each component of the method.
2. Although the motivation for this work is multi-dataset semantic segmentation, the experimental comparison methods do not include the latest methods from the multi-dataset semantic segmentation community, such as [1,2], etc.
3. This paper only uses HRNet-W48 as the backbone. The authors are encouraged to further explore the scalability of the proposed method on transformer-based networks.
4. From Table 5, it appears that the improvement in results due to label description is not significant. However, using GPT to complete this step inevitably incurs a substantial consumption of time and resources, which raises questions about whether it is worth it. Additionally, the authors should discuss the impact of the proposed method on time and space costs.

[1] Gu X, Cui Y, Huang J, et al. Dataseg: Taming a universal multi-dataset multi-task segmentation model. NeurIPS 2023.

[2] Wang L, Li D, Liu H, et al. Cross-dataset collaborative learning for semantic segmentation in autonomous driving. AAAI 2022.

**Questions:**

Please see Weaknesses.

**Limitations:**

Limitations are discussed in the Conclusion.

---

> ### Author Rebuttal · Authors · 2024-08-07
>
> Thank you very much for taking the time to review our work. Below, we summarize each of your questions and provide detailed responses.
>
> **Q1 & Q2: Concerns about the lack of detailed explanation and the omission of the latest methods**
>
> A: We respectfully disagree with your assessment. Our approach is fundamentally different from existing methods and two methods you mention. While existing methods require manual re-labeling (e.g., MSeg [23]), expert knowledge (e.g., NLL+ [4]), or are limited to only two datasets (e.g., Auto Univ. [6]), our method **automatically** constructs **a unified label space** across multiple datasets without human intervention or iterative processes. Furthermore, we are the first to leverage Graph Neural Networks for this task.
>
> The papers you mentioned do not construct a unified label space. Instead, they utilize different segmentation heads (with different weights) for evaluation across various datasets. For instance, [1] employs a text encoder to encode label categories into corresponding embedding spaces for each dataset, predicting within dataset-specific embedding spaces. This method also struggles with handling categories that share the same name but have different annotation standards across datasets (e.g., IDD "curb" vs. MPL "curb"). This challenge is precisely why we introduced $d_i$ in Equation (1), allowing nodes with the same name from different datasets to obtain distinct node features, thereby differentiating these nodes.
>
> On the other hand, [2] uses dataset-specific batch normalization and heads, which can be heavily influenced by dataset biases. In real-world inference scenarios, the outputs from different segmentation heads may conflict, complicating practical application. Our method, however, consistently predicts across different datasets using a unified label space (with the same weights), employing different boolean label mapping matrices solely for performance evaluation. In practical inference, this label mapping is unnecessary. Therefore, methods [1-2] require prior knowledge of the target label space for predictions, which does not align with our task setup and is not suitable for comparison.
>
>
> **Q3: Suggestion to explore scalability on transformer-based networks.**
>
> A: Thank you for your valuable suggestion. We acknowledge the lack of experiments exploring the effectiveness of our method across different models. Training on seven datasets requires approximately one week on four 80G A100 GPUs, which limits our ability to provide results using transformer-based networks at this time. However, we plan to investigate the scalability of our method with transformer-based architectures in future work.
>
>
> **Q4: Concerns about the significance of improvements from label descriptions and resource consumption.**
>
> A: We appreciate your feedback regarding the use of GPT for generating text descriptions. It is important to clarify that the process of generating these descriptions using the text encoder is not conducted in real-time; it involves a one-time inference step. The time cost for this step is less than one minute, and the space requirement for preserving text features is only 3.5MB, which is negligible compared to the overall training costs. Furthermore, after training, the GNN component is discarded, meaning it does not introduce any additional overhead during the inference phase.
>
>
> We hope these clarifications address your concerns effectively.

---

> > ### Comment · Reviewer_dye9 · 2024-08-13
> >
> > I acknowledge the authors' efforts in the rebuttal and have read it. The paper itself is interesting and adequate from a technical point of view. However, my main concern is the lack of comparisons and discussion with previous SOTA in the semantic segmentation community, which hinders the evaluation of this paper. I believe that high time complexity should not be a reason to avoid comparisons. On the contrary, it could introduce new challenges for practical applications. I suggest the authors include Transformer-based comparison methods and optimize the time complexity. Therefore, I will maintain my score.

---

> ### Author Response · Authors · 2024-08-13
>
> Dear Reviewer,
>
> Thank you for your thoughtful feedback and for acknowledging our efforts in the rebuttal. To the best of our knowledge, the SOTA method we refer to is the one presented in IJCV 2024 [5], where our approach demonstrated significant performance improvements in both the 7ds and WD2 ( 51.3 vs. 56.4 on 7ds and 46.9 vs. 50.2 on WD2). Could you please clarify if the SOTA methods you are referring to are those mentioned in your initial review? If so, we have already emphasized that these methods are not directly comparable within our *dataset-agnostic* setting, as they only provide *dataset-aware* prediction and are unable to provide predictions in a unified label space. If you are referring to other methods, we would appreciate it if you could provide some examples for our reference.
>
> We would also like to emphasize that we did not intend to avoid comparisons. Rather, it was challenging to produce the comparison results within the limited time frame. We plan to include additional experimental data in the revised paper, such as results on WD2 using the full RVC collection for training and on 7ds with the initialized adjacency matrix without GNN training. As for your suggestion to include Transformer-based methods, we greatly appreciate this valuable input. We will certainly consider exploring and discussing these approaches in a future version of the paper.
>
> Thanks! We look forward to your any further feedback.

---

> > ### Comment · Reviewer_dye9 · 2024-08-13
> >
> > What I mean by SOTA methods is that they are transformer-based approaches. The proposed method brings considerable improvements with HRNet or SNp as the backbone. However, my main concern is that HRNet and SNp are simple backbones proposed years ago. Therefore, I remain skeptical whether the improvements are still reasonable when the backbone is replaced by more complex and effective ones such as a Transformer. The authors suggested replacing HRNet with other SOTA backbones such as Transfomrer, to validate the generalization and effectiveness of their proposed method.

---

> > > ### Author Response · Authors · 2024-08-14
> > >
> > > Dear Reviewer,
> > >
> > > Thank you for highlighting this important consideration. We fully agree that exploring Transformer-based methods could further validate the generalization and effectiveness of our proposed approach. However, we have currently chosen to use CNN models for the following two reasons, with plans to explore Transformer-based models in future work:
> > >
> > > * For consistency and fairness in comparison with other multi-dataset semantic segmentation training methods, we have utilized CNN models, as these are also employed by the methods we compared against. The consistency of using CNNs allowed us to establish a solid baseline and ensure that our comparisons were on equal footing.
> > >
> > > * As you mentioned, Transformer-based methods have demonstrated significant improvements over traditional CNN backbones like HRNet and SNp. However, we would like to highlight that, even with a CNN backbone, our current method has achieved state-of-the-art results on WD2 benchmark, surpassing other approaches on the leaderboard, including those that employ Transformer-based methods [26][44]. We anticipate that incorporating a Transformer backbone could likely yield even greater improvements.
> > >
> > > We are committed to further enhancing our approach by exploring Transformer-based models in future research, and we appreciate your valuable feedback on this matter.

---

> > > > ### Comment · Reviewer_dye9 · 2024-08-14
> > > >
> > > > I acknowledge the authors' efforts. The authors expect that using Transformer models will lead to greater improvements. However, based on my experience, some methods that are quite effective with CNNs see only limited improvements or even a decline in performance when switched to Transformers. Therefore, I still believe that papers submitted to top machine learning conferences should validate the effectiveness of their methods with Transformers. That said, I am open to discussions with other reviewers and AC. If everyone believes this concern is insignificant, I am inclined to accept the paper. Accordingly, I will lower my confidence rating to 3.

---

### Official Review · Reviewer_M8bi · 2024-07-22

**Soundness:** 3
**Presentation:** 3
**Contribution:** 3
**Rating:** 7
**Confidence:** 4

**Summary:**

This paper introduces a method to automatically match and unify different label spaces for semantic segmentation. This allows to train a single model on multiple, differently annotated datasets. The authors can show that this can yield benefits in overall model performance, also compared to other multi-label approaches. The method is the currently best result on the public WildDash2 benchmark.

**Strengths:**

-	This method is the current SOTA on the public learderboard of WildDash2, which means the proposed method holds against a very rigorous testing setup.
-	With the current knowledge that data scale is one of the key ingredients in well-generalizing models, progress on multi-dataset-training has high potential for impact. For example, the current leader of all ScanNet 3D segmentation leaderboards is also a method based on multi-dataset-training.
-	The presentation is very clear and structured for the most parts of the paper.

**Weaknesses:**

-	It is currently unclear to me based on what information the graph connectivity is learned. How the method is described, the name of the annotated label is the only information that is put into the nodes, and all matching is based on that plus a hallucinated description from GPT. However, the authors mention in line 199-200 that the method would leverage visual similarity between annotations. It is not clear to me how this is facilitated.
-	All results are based on an architecture from 2019. It is lucky that WildDash2 has no stronger competitors in the leaderboard, but it remains unclear whether the method would yield the same improvements for a more competitive base architecture (e.g., mIoU of 77% on Cityscapes is a very bad baseline performance in Table 2. The achieved “improved” 80.7% are on-par with DeepLabv3, which was SOTA in 2018).
-	The method underperforms on more general indoor/outdoor datasets. Both on ADE and COCO, MSeg is better (even when predicting less classes, and in contrast to the bolded numbers, which are somehow ignoring MSeg)
-	I don’t think it is well discussed how hallucinating label descriptions through ChatGPT can introduce mistakes into the methodology. I don’t know about all of the datasets, but for example Cityscapes provides their own descriptions of the labels, which are the descriptions that also label workers use to label the data (see https://www.cityscapes-dataset.com/dataset-overview/#labeling-policy ). LLMs that are simply prompted with the label name can easily make up wrong descriptions that are not calibrated to the label policy of the annotation.

**Questions:**

see above. The points that can hopefully easily be clarified by the authors are the question how visual information is used in the matching, what to do about GPT outputting wrong definitions, and why the method underperforms on more general datasets with larger label spaces.

**Limitations:**

-	The broader impact does not at all discuss potential societal impact of the work. While it is true that it reduces the re-labeling effort, the method still requires segmentation labels, which are the much higher cost compared to re-labeling existing segmentation labels. What should be rather discussed are the societal impact of deploying a method with an automatically generated label space, and what the implications of this are to safety and certification when trying to deploy this method e.g. to automated driving.
- The discussed limitations are OK. It would be interesting of the authors could further comment on ways how to potentially get around the scaling problem of current requiring all datasets loaded on one node, and every dataset beeing sampled for every batch.

---

> ### Author Rebuttal · Authors · 2024-08-07
>
> Thank you for your valuable feedback and for highlighting the areas that require clarification in our paper. We appreciate your insights, which allow us to improve our manuscript. Below, We summarize each of your questions and provide detailed responses.
>
> **Q1: Clarification on how graph connectivity is learned and visual similarity is utilized.**
>
> A: Thank you for your question. The node features in our approach only contain textual characteristics. The visual information, is provided by the segmentation model during training. Specifically, the graph neural network outputs embeddings and adjacency matrix that are utilized in the segmentation network's inference process. For the image training samples, the segmentation network encodes visual embedding features. The GNN makes predictions by classifying and mapping these visual features to labels. The loss function is computed using formulas (4-6), and the loss value is backpropagated through the GNN model to update the parameters.
>
>
> **Q2: Concerns about using an older architecture and its competitive performance.**
>
> A: We appreciate your observations regarding the architecture used in our studies. To ensure a fair comparison, we selected this architecture based on the configuration outlined in Mseg [23]. Training on seven datasets is indeed time-consuming (approximately one week on four 80G A100 GPUs), which limits our capability to evaluate more competitive architectures at this stage. However, our results (Table 11) from five datasets show potential for improvement, as we've enhanced Cityscapes' mIoU to 82.2%. This indicates that there is room for further optimization in our method. We will explore more competitive base architectures, such as transformer-based models, in our future research.
>
>
> **Q3: Performance evaluation relative to the MSeg approach on indoor/outdoor datasets.**
>
> A: Thank you for raising this question. It's worth noting that MSeg combines categories and overlooks some difficult classes, which simplifies the learning task. Many of these difficult classes correspond to IoU values lower than the overall mean IoU (e.g., for ADE, mIoU for omitted/merged classes is 37.5 vs. all categories at 42.0; for COCO, it's 37.1 vs. 46.7). MSeg's merged categories also take into account label alignment across different datasets, which eases the learning process and thus may not represent a fair comparison to our method. We have provided examples of merged categories and their IoUs to help illustrate this point:
>
> |MSeg Label| ADE Label| IoU|
> |---|--|--|
> | **table**| coffee table| 55.9|
> || table| 50.6|
> | **terrain**| land| 0.4|
> || earth| 31.1|
> || field| 22.1|
> || grass| 64.7|
> || sand| 36.4|
> || dirt track| 6.4|
> | **mountain_hill**| mountain| 56.2|
> || hill| 8.9|
> | **car**| car| 84.3|
> || van| 35.9|
> | **stairs**| stairs| 28.5|
> || stairway| 28.9|
> | **railing_banister**| railing| 26.4|
> || bannister| 10.0||
> | **unlabeled**| computer| 66.3|
> || signboard| 36.3|
> || monitor| 45.5|
> || crt screen| 5.0|
> || screen| 63.9|
> || canopy| 8.8|
> || plant| 44.0|
> || bar| 23.8|
> || step| 11.7|
>
> | MSeg Label| COCO Label| IoU|
> | --| --|--|
> | **building**| house| 30.0|
> || roof| 52.7|
> || building-other-merged| 46.6|
> | **floor**| floor-wood| 36.0|
> || floor-other-merged| 47.8|
> | **table**| table-merged| 38.3|
> || dining table| 75.8|
> | **terrain**| grass-merged| 37.6|
> || sand| 14.5|
> || dirt-merged| 22.5|
> | **wall**| wall-brick| 24.9|
> || wall-stone| 54.9|
> || wall-tile| 30.3|
> || wall-wood| 16.0|
> || wall-other-merged| 43.1|
> | **vegetation**| flower| 10.9|
> || tree-merged| 49.5|
>
>
> **Q4: Potential issues with hallucinated label descriptions through ChatGPT.**
>
> A: Thank you for your invaluable input regarding the potential inaccuracies in generated label descriptions. We agree that using official label descriptions from datasets like Cityscapes greatly enhances the reliability of our methodology. Given that some datasets may lack corresponding label descriptions or have inconsistent language styles, this can pose challenges for the model. Therefore, in the future, we plan to use the label descriptions provided by official datasets as prompts, employing GPT models to generate more accurate and stylistically consistent label descriptions to help the model learn better. This adjustment may help improve the model's learning process by reducing the risks associated with hallucinated descriptions.
>
>
> **Limitation: Discussion on societal impacts and scalability challenges.**
>
> A: Thank you for the addition to the limitations section. Indeed, compared to unsupervised or weakly supervised methods, we still require complete annotated data. However, our approach leverages the available annotation information in existing datasets to reduce the reliance on annotated data. Errors in the fully automated construction of a unified label space do present some safety risks for autonomous driving tasks. Therefore, we also recommend introducing a manual review mechanism to address these issues. Current automatic label generation methods may introduce significant safety risks, so we recommend incorporating a manual review mechanism for generated labels to ensure accuracy and mitigate these concerns.
>
> Regarding scalability, we concur that it's not necessary to load all datasets on a single node. Instead, as long as the computations for weight gradient updates include representations from all datasets, we can distribute this across multiple nodes, allowing more efficient processing. This approach can help facilitate scaling while avoiding bottlenecks in performance. We will include additional descriptions of these limitations in the final version of the paper, particularly focusing on safety considerations for autonomous driving scenarios.
>
> We hope that these responses provide the clarity you were seeking and address your concerns adequately. We sincerely appreciate your constructive feedback, which is invaluable in improving our manuscript.

---

> > ### Comment · Reviewer_M8bi · 2024-08-12
> > **read rebuttal**
> >
> > Dear authors,
> >
> > thank you for your rebuttal, which I have read. Ultimately my view on this paper does not change. There seems to be a limitation of hallucinating definitions for labels and matching them purely on textual information, where visual information for matching is only propagated during training, meaning unseen datasets are aligned purely on GPT-generated definitions.
> >
> > I don't fully understand the comment about MSeg. Are the authors suggesting mIoU values are compared between methods that are measured over different sets of labels? That would be strange. In any case, I can accept that MSeg does not follow a standard protocol for these datasets.
> >
> > Overall I lean towards keeping my original score of accept, pending the discussion with other reviewers.

---

> ### Author Response · Authors · 2024-08-13
>
> Dear Reviewer,
>
> Thank you for your thoughtful feedback. We acknowledge the limitation of our current approach, where label descriptions generated by GPT are used to construct node features purely based on textual information. In future work, we plan to address this limitation by incorporating official label descriptions and introducing mask image features corresponding to the label categories to enhance the node features.
>
> Regarding your question about MSeg, we would like to clarify that MSeg performs label merging or omission for certain datasets (e.g., ADE, COCO), while for others (e.g., CS, SUN), it uses the same evaluation categories as we do. For these datasets where a different evaluation protocol was used, while direct comparisons are not possible, this does explain the higher mIoU values observed in their results. However, we believe that on datasets where the same evaluation protocol is used, a fair comparison can be made.
>
> Thank you again for your detailed feedback and for maintaining your positive view of our work.

---

### Official Review · Reviewer_kdwE · 2024-07-22

**Soundness:** 2
**Presentation:** 3
**Contribution:** 3
**Rating:** 5
**Confidence:** 5

**Summary:**

The paper presents an approach to automatically train a semantic segmentation network using multiple datasets with individually different class label policies. A unified label space emerges during training automatically without direct manual label mapping definitions by using a graph neural network (GNN) which is guided by textual descriptions of each label in unison with a multi-head segmentation head. The training alternates between GNN and segmentation network training to jointly optimize the label mapping for multiple dataset as well as the actual image segmentation task itself. Multiple experiments evaluate the approach to unify 7 mixed datasets (CS, MPL, SUN, BDD, IDD, ADE, COCO) as well as 5 road-scene datasets (CS, IDD, BDD, MPL, WD2). Performance is compared by both training single heads and using the Multi-Seg. Head. as well as benchmarking on WD2. Additional ablation experiments showcases the value of textual label descriptions for the unification process.
In summary this approach allows for a joint training with multiple mixed segmentation datasets without additional manual intervention.

**Strengths:**

*) Generation of segmentation label space mappings to automatically create a unified label space.
*) Great overall performance for each tested dataset
*) Record benchmark results for Wilddash2
*) Many experiments to showcase the approach on multiple driving-scene datasets as well as some more generic datasets.
*) Supplemental contains the full training code to recreate the experiments

**Weaknesses:**

*) Some details are omitted (see Questions) making it hard to judge the extent/validity of some claims (e.g. robustness/generalization).
*) Multi-head training and evaluations favor the accumulation of per-dataset bias. The cross-validations are a good start but there is no evaluation of "new"/out-of-distribution labels from one dataset within the other.  This amplifies dataset-specific training (each head is optimized to work for one specific dataset); thus training to "solve" datasets rather than the task at hand. L281 (tunnel == fireplace) is a great observation but also shows the weakness of the approach.
*) "Splitting" of source labels is not supported; the multi-heads are trained in isolation so if a label in one dataset requires splitting in another dataset, the respective heads are isolated (one has the split, the other doesn't). During inference of an unseen image this can result in only a single merged representation instead of the more detailed split. This effect is underrepresented during evaluation as dataset bias further separates the heads to work in favor of the overlapping test data.
*) Both WD1 and WD2 are used. WD2 supersedes WD1 in every way /(it also includes all WD1 frames with the extended label policy) and crucially also uses a unified label policy combining IDD, MPL, CS & WD1 (80 label cat.; the benchmark uses the reduced WD2_eval 25 cat policy). So contrary to L223, WD2 does contain the labels "lane marking", "crosswalk" and "manhole". WD2 has a different publication associated with it which is unmentioned: CVPR22 "Unifying Panoptic Segmentation for Autonomous Driving". The paper/experiments would be easier to understand with only WD2.
*) For some figures/descriptions in the paper it is unclear which dataset combinations was used during training. Introduction of fixed names (e.g. 5ds vs. 7ds for 5/7 datasets; or explicitly mention "domain-general"/"domain-specific" wording) could help.

**Questions:**

*) The creation of textual descriptions using ChatGPT is not described at all; which version of ChatGPT was used? How were per-dataset label spaces described as textual input? Were images/masks/masked images used with a multi-modal version of ChatGPT (e.g. GPT4V; GPT4o; ...). Using masks together with images as inputs for ChatGPT is non-trivial. L110 contains no details; wording in Figure 2 suggests a multi-modal model as "image of" is part of the dataset label text or simply the "blind" pure-text completion of prompts in the style "An image of <label> from the dataset <dataset>....". Please clarify; the text label descriptions are an important aspect of the paper's novelty!
*) Where is the final resulting unified label mapping?  Figure 5 shows a small part but the full joined label set would give a good indication of generalization vs. specialization.
*) Supplem. B. (target number of unified labels) is a crucial component as it directly steers the separation/merging of similar concepts (e.g. seperate "Cityscapes_road"/"MPL_road" labels vs. a unified "road" label). How is the hyperparameter Lambda|L| chosen and what number of labels did the experiments end up with?
*) What pre-training was used for the initialization of the multi-heads?
*) WD2 negative examples are a defined subset within the benchmarking dataset. However, the official rules of the benchmark state, that both regular/"positive" and negative frames/subsets have to be treated equally. The comparability within the benchmark is only held upright, as long as this rule is observed.  L246/L247 can be interpreted otherwise: "We map non-evaluated classes to a void label"; is this applied to both positive and negative frames alike?

**Limitations:**

*) Limitations relating to dataset bias could be mentioned.

---

> ### Author Rebuttal · Authors · 2024-08-07
>
> We sincerely appreciate your thoughtful inquiries and for pointing out crucial elements of our methodology. Your feedback is instrumental in refining our paper and addressing any weaknesses. Below, we summarize your questions and provide detailed responses.
>
> **Q1: About the creation of textual descriptions using ChatGPT.**
>
> A: Thank you for pointing this out. We apologize for not providing specific details in our initial submission. In the revised version of the paper, we will clarify that we used ChatGPT 3.5 to generate label descriptions without any image input. Each dataset label category was formatted into the text input as “An image of <label> from the dataset <dataset>.” This prompt was used to encourage ChatGPT to complete the label descriptions. The resulting complete label descriptions were then inputted into a text encoder to generate the corresponding textual features for the label nodes.
>
>
> **Q2: Where is the final resulting unified label mapping?**
>
> A: Please refer to our global rebuttal for more examples. We plan to present our complete unified label mapping in the form of open-source code.
>
>
> **Q3: The importance of Supplement B regarding unified labels and the choice of hyperparameter Lambda |L|.**
>
> A: We appreciate you highlighting the importance of this aspect. Based on experimental insights and referencing paper [52], we selected the hyperparameter ($\lambda$ = 0.5), as this setting provided a good initial value. In our experiments with seven training datasets (CS, MPL, SUN, BDD, IDD, ADE, COCO), we had a total of 448 label categories. With the selected parameter, the resulting unified label space \(|L|\) comprised 231 labels. We will include these details in the revised version of the paper.
>
>
>
> **Q4: What pre-training was used for the initialization of the multi-head models?**
>
> A: In the Multi-SegHead Training Stage section of Supplementary A, we describe the process of training our multi-head model. Specifically, we construct a separate segmentation head for each dataset, where each head contains only a linear layer to map the embedded feature channels to the specific label categories of that dataset. The training was conducted using four 80G A100 GPUs for 100K iterations. Image preprocessing and hyperparameter settings were consistent with those used in other training stages.
>
>
> **Q5: Treatment of WD2 negative examples in benchmarking datasets.**
>
> A: We apologize for the lack of clarity in our description, which may have caused confusion. We adhere strictly to the WD2 evaluation guidelines, applying uniform label mapping to all samples. This process is indeed applied to both the positive and negative frames. We will explicitly include this description in the revised version of the paper.
>
> ---
>
> **Weakness 1: Multi-head training and dataset bias concerns.**
>
> A: Thank you for highlighting the issue of dataset bias in multi-head training. We acknowledge this limitation. After completing cross-validation, we discard the multi-heads and the model utilizes a unified label embedding space for predicting class probabilities during subsequent training phases, which should help mitigate biases introduced by the initial training setup. We believe adding more datasets may help alleviate out-of-distribution label concerns, as combining datasets may reduce errors from mislabeling during cross-validation. We also recognize the importance of distinct semantic gaps, such as between "tunnel" and "fireplace". We plan further research to leverage label descriptions from official datasets and utilize advanced large models to provide accurate label mappings while constraining the model’s output based on textual features to minimize incorrect connections between semantically different classes.
>
>
> **Weakness 2: Splitting of source labels with respect to multi-head training.**
>
> A: We apologize for any confusion regarding the isolation of multi-heads training. After the initial multi-seghead training stage, we discard the multi-head structure, and thus, only a single UniSeghead is maintained for predictions. During inference on unseen images, we solely output results from the unified label space, eliminating the complications related to the isolation of respective heads. This will be clarified in the revised version of the paper. Additionally, our visualization results (in Supplement F) show that the final model generates fine-grained segmentation results.
>
>
> **Weakness 3 & 4: The use of both WD1 and WD2, and unclear figures or descriptions.**
>
> A: Thank you for noting the potential confusion arising from using both WD1 and WD2. To clarify, our model did not utilize WD1 during training, and we plan to remove references to testing on WD1 in the final version of the paper for better understanding. We also appreciate your reminder regarding the citation of the WD2 paper, which we will include in the revised version of the paper. Additionally, thank you for pointing out the lack of model descriptions in our visual results that caused confusion. The L223 experiment in the paper refers to the 7ds training model, which does not include WD2, resulting in only MPL having *lane markings*, *crosswalks*, and *manholes*. We will add more detailed descriptions to the images (Fig. 3 and Fig. 5) and revise 'Ours' to 'Our 7ds model'."
>
>
> We hope these responses provide clarity and address your concerns effectively. Thank you once again for your constructive feedback.

---

### Official Review · Reviewer_GmR6 · 2024-07-23

**Soundness:** 3
**Presentation:** 3
**Contribution:** 3
**Rating:** 6
**Confidence:** 4

**Summary:**

This paper presents a novel approach for automated label unification across multiple datasets in semantic segmentation using Graph Neural Networks (GNNs). The proposed method aims to create a unified label space that allows semantic segmentation models to be trained on multiple datasets simultaneously without the need for manual re-annotation or taxonomy reconciliation. The results demonstrate significant performance improvements over existing methods, achieving state-of-the-art performance on the WildDash 2 benchmark.

**Strengths:**

+ Novel Approach: The use of GNNs to automatically construct a unified label space is innovative and addresses the challenge of conflicting label spaces in multi-dataset training.
+ Performance: The method shows impressive performance improvements, achieving state-of-the-art results on the WildDash 2 benchmark and outperforming other multi-dataset training methods.
+ Efficiency: The approach eliminates the need for manual reannotation and iterative training procedures, significantly enhancing the efficiency of multi-dataset segmentation model training.
+ Robustness: The method demonstrates good generalization to unseen datasets, indicating its robustness and applicability across various scenarios.

**Weaknesses:**

This indeed seems a interesting problem. I have several questions:

1. How do you select the unified label size N? Is it adaptive to different dataset labels, or preset hyper-parameters?
2. Can you show more results of the bipartite graph generated the GNN? Like what labels is hard to link by text features, but linked by the GNN.
3. Is the generated graph by GNN influenced by the number of datasets in training or the models? For example, if I use different models for the training, will the graph be the same? If I use two dataset in multi-dataset training, will the nodes of these two dataset labels have the same edges as in when I use three datasets?
4. Can I apply the generated graph to other models' training without the need for doing GNN training again? If not, how much training cost will GNN bring?

**Questions:**

Please see the weaknesses.

**Limitations:**

The paper is interesting and the experiments seem to prove the effectiveness. But I need more evidence to prove the robustness of the GNN results when using different datasets and models.

---

> ### Author Rebuttal · Authors · 2024-08-07
>
> We sincerely thank for your thoughtful questions. Below are our responses to the specific questions raised:
>
> **Q1: How do you select the unified label size N? Is it adaptive to different dataset labels, or preset hyper-parameters?**
>
> A: We apologize for not clearly explaining the selection of the unified label size N in the main text, which might have caused some confusion. In Supplementary Material B, we clarify that N is determined by solving Algorithm 2, rather than being a preset hyper-parameter. Specifically, we construct the label co-occurrence relationships through cross-validation of the multi-seghead model across different datasets. This automatic solving method allows us to adapt to different dataset labels and enhances scalability. We will add this explanation to Section 3.2 of the main text in the revised version of the paper.
>
> **Q2: Can you show more results of the bipartite graph generated by the GNN? Like what labels are hard to link by text features, but linked by the GNN.**
>
> A: Please refer to our global rebuttal for more examples. For instance, ADE: Signboard is annotated as both Traffic sign (front) and Traffic sign (back), while CS annotates Traffic sign (back) as an ignore category. We believe that such knowledge should be learned through samples.
>
> **Q3: Is the generated graph by GNN influenced by the number of datasets in training or the models? For example, if I use different models for the training, will the graph be the same? If I use two datasets in multi-dataset training, will the nodes of these two dataset labels have the same edges as when I use three datasets?**
>
> A: Yes, the selection of different models and datasets affects the constructed nodes. The GNN relies on the model’s segmentation predictions to learn the label mapping matrix. If the model cannot accurately recognize fine-grained categories, the GNN will not be able to construct the label mappings for these fine-grained categories. The influence of datasets on the generated graph is likely present. For instance, in 5ds training, IDD: Non-drivable fallback or rail track is merged with Terrain. However, in 7ds training, after removing WD2: Terrain and adding COCO: Railroad, IDD: Non-drivable fallback or rail track is merged with the fine-grained categories MPL: Rail Track and COCO: Railroad. We speculate this is because the inclusion of COCO: Railroad in 7ds training improves the model’s learning of fine-grained categories, affecting the prediction results for Non-drivable fallback or rail track in the IDD dataset, leading to different outcomes.
>
> **Q4: Can I apply the generated graph to other models' training without the need for doing GNN training again? If not, how much training cost will GNN bring?**
>
> A: Yes, similar to other multi-dataset training methods that apply manually designed unified label spaces to different models, our GNN model constructs a unified label space as a boolean label mapping matrix that can be separated and used for training other models. We will open-source the constructed unified label space along with the revised version of our code to facilitate training with other models.
>
> If there are any further questions or concerns, please feel free to reach out to us.

---

> > ### Comment · Reviewer_GmR6 · 2024-08-13
> >
> > Thanks for the response. The rebuttal is clear and solves my concern. I would like to raise my rating to 6.

---

### Author Rebuttal · Authors · 2024-08-07

We sincerely thank all the reviewers for their valuable comments and suggestions. The automatic construction of a unified label space using GNNs is the main innovation and contribution of our method. However, due to the limitations of our presentation format, we apologize for not being able to provide a comprehensive demonstration. We plan to open-source the constructed unified label space along with our code. In the supplementary material, we also present more examples of the generated graphs.

Our supplementary material is divided into four parts:

1. **Fig. 1**: Shows well-constructed examples across the 7 datasets.
2. **Fig. 2**: Illustrates examples of the label space initialized using Algorithm 2 and subsequently refined through GNN training.
3. **Fig. 3**: Displays examples of some incorrectly constructed links.
4. **Fig. 4**: Shows the label space constructed on 5 datasets.

Since our method constructs the unified label space through sample-based learning, the constructed label space depends on the samples included in the datasets. For instance, in **Fig. 2**, Traffic sign (back) is labeled as an ignored category in CS, BDD, and IDD, making it difficult to learn a link with MPL: Traffic sign (back). Additionally, the model's learning relies on visual features. For example, COCO: Window-other often includes vegetation outside the window, leading to an incorrect link in **Fig. 3**. Similarly, the tunnel == fireplace issue mentioned in the paper arises because these categories have similar visual features. The model merges them into one category to save space for predictions. However, these categories have significant semantic differences. We believe that using label descriptions provided by official datasets to generate more accurate text features, combined with incorporating knowledge from large models, can help constrain the model’s label links and reduce incorrect semantic links. This is an area worth further research.

We thank all the reviewers for your insightful questions and feedback. If there are any more issues or concerns to discuss, we welcome further inquiries and are happy to provide additional clarifications.

---

### Comment · Reviewer_kdwE · 2024-08-09

Dear Authors,

thanks for the great work and the clarifications/ additional efforts you put into the rebuttal!
In general I'm satisfied with the update and would currently vote for Accept.
The main weakness/flaw that remains is the hallucinated description of the labels generated by GTP3.5 solely from essentially extending a text "<label> for dataset <name>" without incorporating the images/masks themselves. However, it is interesting/provocative that this improves the automatic label unification. The results/generalizations are promising and the approach is novel enough.

Thanks!

---

> ### Author Response · Authors · 2024-08-12
>
> Dear Reviewer,
>
> thank you very much for your positive feedback. We acknowledge your concern regarding the hallucinated descriptions of the labels generated by GPT-3.5, particularly the potential shortcomings of relying on extended text without incorporating the images or masks themselves. We agree that integrating the corresponding image masks into the node features could enhance the accuracy of label generation and help mitigate the issues related to erroneous outputs from the model. Indeed, this is an important consideration, and we plan to explore this direction in our future research.
>
> Thanks!

---

### Decision · Program_Chairs · 2024-09-25

**Decision:**

Accept (poster)

**Comment:**

This manuscript addresses the learning of a unified class space given several datasets and their particular labeling conventions. Such approaches need to recover the incidence matrices M_d that link universal classes with the corresponding labels of the dataset d. Different than in the previous work, the proposed method recovers soft incidence by a GNN that operates upon the bipartite graph that associates textual label embeddings with the universal visual vectors. The GNN aligns the universal predictions of a frozen feature extractor with the labels and encourages orthogonality of the universal visual vectors.

The manuscript has received in-depth reviews and widely divergent ratings. The diversity of viewpoints reflects the complexity of this multifaceted problem. The reviewers commend the joint recovery of universal class embeddings and dataset-to-universal incidence, as well as state of the art performance on the WildDash dataset. The criticism can be subdivided into low-level and high-level issues. The low-level issues involve limited impact of textual label embeddings due to relying on chatgpt descriptions, theoretically unfounded loss (log-softmax-sum vs log-sum-softmax), missing comparison with previous work [5], and outdated baselines. Most of these issues could be addressed in a straight-forward manner.

High-level criticism addresses vagueness of the chosen performance metric (mIoU). Ultimately, we can not say whether the performance is great due to better universal mappings or more aligned training setup and more GPU RAM. It appears that an additional performance metric (eg AP [40]) would be helpful in this regard as well as validation of the GNN capacity and more entries in Table 5 (eg independently recovering the incidence and the universal vectors, without using GNN).

Ultimately, I agree with M8bi. This paper can be both rejected due to insufficient technical correctness, and accepted due to significance of the research problem.